# Learning Similarity Metrics for Numerical Simulations

## Abstract

We propose a novel approach to compute a stable and generalizing metric (*LNSM*) with convolutional neural networks (CNN) to compare field data from a variety of numerical simulation sources. Our method employs a Siamese network architecture that is motivated by the mathematical properties of a metric and is known to work well for finding similarities of other data modalities. We leverage a controllable data generation setup with partial differential equation (PDE) solvers to create increasingly different outputs from a reference simulation. In addition, the data generation allows for adjusting the difficulty of the resulting learning task. A central component of our learned metric is a specialized loss function, that introduces knowledge about the correlation between single data samples into the training process. To demonstrate that the proposed approach outperforms existing simple metrics for vector spaces and other learned, image based metrics we evaluate the different methods on a large range of test data. Additionally, we analyze generalization benefits of using the proposed correlation loss and the impact of an adjustable training data difficulty.

## 1 Introduction

Evaluating computational tasks for complex data sets is a fundamental problem in all computational disciplines. Regular vector space metrics, such as the $L^2$ distance were shown to be very unreliable (Wang et al., 2004; Zhang et al., 2018), and the advent of deep learning techniques with convolutional neural networks (CNNs) made it possible to more reliably evaluate complex data domains such as natural images, texts (Benajiba et al., 2018), or speech (Wang et al., 2018). Our central aim is to demonstrate the usefulness of CNN-based evaluations in the context of numerical simulations. These simulations are the basis for a wide range of applications ranging from blood flow simulations to aircraft design. Specifically, we propose a novel learned numerical simulation metric (*LNSM*) that allows for a reliable similarity evaluation of simulation data.

Potential areas of application for a such a metric are fundamental problems that arise in all areas where such numerical simulations are performed: among others, accuracy evaluations of existing and new simulation methods with respect to a known ground truth solution (Oberkampf et al., 2004) could be performed more reliably than with a regular vector norm. In addition, the energy landscape for parameter inference via optimization through a differentiable solver could be improved in order to match experimental data or observations, and generative networks for physical phenomena such as GANs, could employ a specialized metric to replace commonly used perceptual loss terms.

In this work, we focus on field data, i.e. dense grids of scalars, similar to images, which were generated with known partial differential equations (PDEs) in order to ensure the availability of ground truth solutions. While we focus on 2D data in the following to make comparisons with existing techniques from imaging applications possible, our approach naturally extends to higher dimensions. Every sample of this 2D data can be regarded a high dimensional vector, so metrics on the corresponding vector space are applicable to evaluate similarities. These metrics are typically simple, elementwise functions such as $L^1$ or $L^2$ distances. We denote these as shallow metrics, and their inherent problem is that they can not capture structures of any scale or contextual information.

This problem is not confined to the field of natural images: it represents a fundamental challenge for many areas of simulation, most prominently for the field of turbulence simulations (Moin & Mahesh, 1998; Lin et al., 1998). Many practical problems require solutions over time, and need a

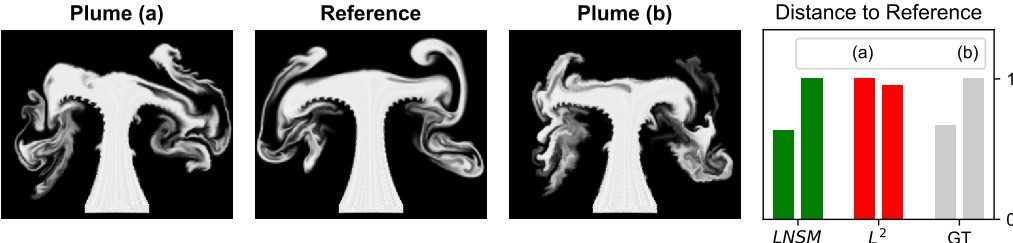

**Figure 1:** Example of field data from a fluid simulation of hot smoke with normalized distances for different metrics. Our method (*LNSM*, in green) approximates the ground truth distances (GT, gray) determined by the data generation method best, i.e., version (a) is closer to the ground truth data than (b). An $L^2$ metric (red) erroneously yields a reversed ordering.

vast number of non-linear operations that often result in substantial changes of the solutions even for small changes of the inputs. Hence, despite being based on known, continuous formulations, these systems can be seen as *chaotic*. We illustrate this behavior in Fig. 1, where two smoke flows are compared to a reference simulation. A single simulation parameter was varied for these examples, and a visual inspection shows that smoke plume (a) is more similar to the reference. This matches the data generation process: version (a) has a significantly smaller parameter change than (b), as shown in the inset graph on the right. Our *LNSM* robustly predicts the ground truth distances, while the $L^2$ metric labels plume (b) as more similar. In our work, we focus on retrieving the relative distances of simulated data sets. Hence, we do not aim for retrieving the absolute parameter change, but a relative distance that preserves ordering with respect to this parameter.

Using existing image metrics based on CNNs for this problem is not optimal either: Natural images only cover a small fraction of the space of possible 2D data, and numerical simulation outputs are located in a fundamentally different data manifold within this space. Hence, there are crucial aspects that can not be captured by purely learning from photographs. Furthermore, we have full control over the data generation process for simulation data. As a result, we can create arbitrary amounts of training data with gradual changes and a known ground truth order. With this data, we can learn a metric that is not only able to directly extract and use features, but additionally knows the fundamental interactions between them. The central contributions of our work are:

- A Siamese network architecture with feature map normalization which is able to learn a metric that generalizes well to unseen simulations methods.

- We also propose an improved loss function that combines a batchwise correlation loss term with a mean squared error to improve the accuracy of the learned metric.

- In addition, we show how a data generation approach for numerical simulations can be employed to train networks with general and robust feature extractors for metric calculations.

## 2 RELATED WORK

One of the earliest methods to go beyond using simple metrics based on $L^p$-norms for natural images was the structural similarity index (Wang et al., 2004). Despite improvements, this method can still be considered a shallow metric. Over the years multiple large databases for human evaluations of natural images were presented, for instance CSIQ (Larson & Chandler, 2010), TID2013 (Ponomarenko et al., 2015), and CID:IQ (Liu et al., 2014). With this data and the discovery that CNNs can create very powerful feature extractors that are able to recognize patterns and structures, deep feature maps quickly became established as means for evaluation (Amirshahi et al., 2016; Berardino et al., 2017; Bosse et al., 2016; Kang et al., 2014; Kim & Lee, 2017). Recently, these methods were improved by predicting the distribution of human evaluations instead of directly learning distance values (Prashnani et al., 2018; Talebi & Milanfar, 2018b). Finally, Zhang et al. (2018) compared different architecture and levels of supervision, and showed that metrics can be interpreted as a transfer learning approach, by applying a linear weighting to the feature maps of any network architecture to form the image metric *LPIPS v0.1*. Typical use cases of these image-based CNN metrics are computer vision tasks like detail enhancement (Talebi & Milanfar, 2018a), style transfer, and super-

resolution (Johnson et al., 2016). Generative adversarial networks also leverage CNN-based losses by training a discriminator network in parallel to the generation task (Dosovitskiy & Brox, 2016).

Siamese network architectures are known to work well for a variety of comparison tasks such as audio (Zhang & Duan, 2017), satellite images (He et al., 2019) or finding similar interior product designs (Bell & Bala, 2015). They were additionally demonstrated to yield robust object trackers (Bertinetto et al., 2016), algorithms for image patch matching (Hanif, 2019), and for descriptors for fluid flow synthesis (Chu & Thuerey, 2017). Inspired by these works we use a similar Siamese neural network architecture for our metric learning task. In contrast to other work on self-supervised learning that utilizes spatial or temporal changes to learn meaningful representations (Agrawal et al., 2015; Wang & Gupta, 2015), our method does not rely on tracked keypoints in the data.

While correlation terms have been used for learning joint representations by maximizing correlation of projected views (Chandar et al., 2016), and are popular for style transfer applications via the Gram matrix (Ruder et al., 2016), they were not used for learning distance metrics. As we demonstrate below, they can yield significant improvements in terms of the inferred distances.

Similarity metrics for numerical simulations are a topic of ongoing investigation. A variety of specialized metrics have been proposed to overcome the limitations of $L^p$-norms, such as the displacement and amplitude score from the area of weather forecasting (Keil & Craig, 2009), and permutation based metrics for energy consumption forecasting (Haben et al., 2014). Turbulent flows, on the other hand, are often evaluated in terms of aggregated frequency spectra (Pitsch, 2006). Crowd-sourced evaluations based on the human visual system were also proposed to evaluate simulation methods for physics-based animation (Um et al., 2017), and for comparing non-oscillatory discretization schemes (Um et al., 2019). These results indicate that visual evaluation methods in the context of field data are possible and robust, but they require extensive (and potentially expensive) user studies. Additionally, our method naturally extends to higher dimensions, while human evaluations inherently rely on projections with at most two spatial and one time dimension.

## 3 Constructing a CNN-based Evaluation Metric

In the following, we explain our considerations when employing CNNs as evaluation metrics. For a comparison that corresponds to our intuitive understanding of how distances work, an underlying *metric* has to obey certain criteria. More precisely, a function $m : \mathbb{I} \times \mathbb{I} \to [0, \infty)$ is a metric with respect to its input space $\mathbb{I}$, if it satisfies the following properties $\forall \boldsymbol{x}, \boldsymbol{y}, \boldsymbol{z} \in \mathbb{I}$:

$$m(\boldsymbol{x}, \boldsymbol{y}) \geq 0 \qquad \text{non-negativity} \qquad (1)$$
$$m(\boldsymbol{x}, \boldsymbol{y}) = m(\boldsymbol{y}, \boldsymbol{x}) \qquad \text{symmetry} \qquad (2)$$
$$m(\boldsymbol{x}, \boldsymbol{y}) \leq m(\boldsymbol{x}, \boldsymbol{z}) + m(\boldsymbol{z}, \boldsymbol{y}) \qquad \text{triangle inequality} \qquad (3)$$
$$m(\boldsymbol{x}, \boldsymbol{y}) = 0 \iff \boldsymbol{x} = \boldsymbol{y} \qquad \text{identity of indiscernibles} \qquad (4)$$

The properties (1) and (2) are crucial as distances should be symmetric and have a clear lower bound. Eq. (3) ensures that direct distances can not be longer than a detour. Property (4), on the other hand, is not really useful for discrete operations as approximation errors and floating point operations can easily lead to a distance of zero for slightly different inputs. Hence, we focus on a relaxed, more meaningful definition $m(\boldsymbol{x}, \boldsymbol{x}) = 0$ which leads to a so called *pseudometric*. It allows for a distance of zero for different inputs, but has to be able to spot identical inputs.

We realize these requirements for a pseudometric with an architecture that follows popular perceptual metrics such as *LPIPS*: The activations of a CNN are compared in latent space, and accumulated with a set of weights. Afterwards, the resulting per-feature distances are aggregated to produce a final distance value. Fig. 2 gives a visual overview of this process.

**Base Network**  The sole purpose of the base network is to extract feature maps from both inputs. The Siamese architecture implies that the weights of the base network are shared for both inputs, meaning all feature maps are comparable. In addition, this ensures the identity of indiscernibles because for identical feature maps a distance of zero is guaranteed by the following operations. In the last years, a variety of powerful CNN-based feature extraction architectures were proposed. We experimented with various networks, such as AlexNet (Krizhevsky et al., 2017), VGG (Simonyan & Zisserman, 2015), SqueezeNet (Iandola et al., 2016), and a fluid flow prediction network (Thuerey

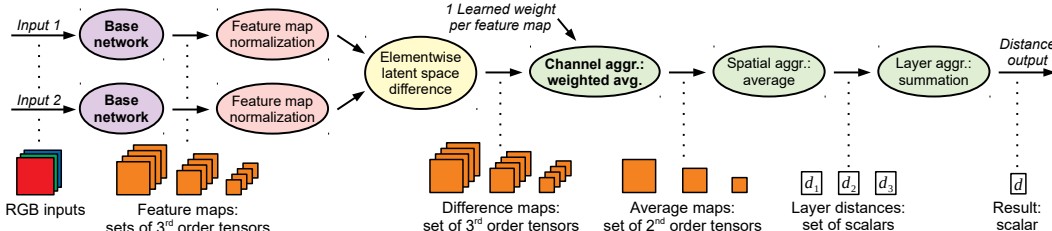

**Figure 2:** Overview of the proposed distance computation for a simplified base network that contains three layers with four feature maps each in this example. The output shape for every operation is illustrated below the transitions in orange and white; bold operations are learned by the CNN.

et al., 2018). In all cases, only the feature extracting layers are used, and the remaining layers which are responsible for the original task, e.g., classification, are discarded. Building on this previous work, we consider three variants of the networks below: using the original pre-trained weights, fine-tuning them, or re-training the full networks from scratch. In contrast to typical CNN tasks where only the result of the final output layer is further processed, we make use of the full range of extracted features across the layers of a CNN (see Fig. 2). This implies a slightly different goal: while early features should be general enough to allow for extracting more complex features in deeper layers, this is not their sole purpose. Rather, features in earlier layers of the network can directly participate in the final distance calculation, and can yield important cues. As we will demonstrate below, we achieved the best performance for our data sets using a custom architecture with five layers, similar to a reduced AlexNet, that was trained from scratch (see App. B.1).

**Feature Map Normalization**   The goal of normalizing the feature maps is to transform the extracted features of each layer, that typically have very different orders of magnitude, into comparable ranges. While this task could potentially be performed by the learned weights, we found the normalization to yield improved performance in general (see App. B.2). Zhang et al. (2018) proposed a length unit normalization using a division by the Euclidean norm in channel dimension, to only measure the angle between the latent space vectors with a cosine distance. Instead, we suggest to interpret all possible feature maps as a normal distribution and to normalize them to a standard normal distribution. This is achieved via a preprocessing step using the full training data set: we subtract the mean of the feature maps and then divide by their standard deviation in channel dimension for each layer. As a result, we can measure angles for the latent space vectors and compare their magnitude in the global length distribution.

**Latent Space Differences**   Combining the latent spaces representations $\tilde{\boldsymbol{x}}, \tilde{\boldsymbol{y}}$ that consist of all extracted features from the two inputs $\boldsymbol{x}, \boldsymbol{y}$ lies at the core of the metric computation. Here, the most obvious approach to employ an elementwise $\tilde{\boldsymbol{x}}_i - \tilde{\boldsymbol{y}}_i$ difference is not advisable, as this would directly violate the metric properties above. Instead, possible options to ensure non-negativity and symmetry are $|\tilde{\boldsymbol{x}} - \tilde{\boldsymbol{y}}|$ or $(\tilde{\boldsymbol{x}} - \tilde{\boldsymbol{y}})^2$. We found that both work equally well in practice. Considering the importance of comparing the extracted features, the simple operations used for comparing the features do not seem optimal. Rather, one can imagine that improvements in terms of comparing one set of feature activations could lead to overall improvements for derived metrics. Hence, we experimented with replacing these operations with a pre-trained CNN-based metric for each feature map. This creates a recursive process, and a "meta-metric" that reformulates the initial problem of the similarity between inputs in terms of the similarity a series of deep representations of the inputs. However, as detailed in App. B.3, we have not found this recursive approach to yield any substantial improvements. This implies that once a large enough number of expressive features is available for comparison, the in-place difference of each feature is sufficient to compare two inputs. In the following, we compute the feature difference maps (Fig. 2, in yellow) via $(\tilde{\boldsymbol{x}} - \tilde{\boldsymbol{y}})^2$.

**Aggregations**   The subsequent aggregation operations (Fig. 2, in green) are applied to the difference maps to compress the contained per feature differences along the different dimensions into a single distance value. The aggregation operations only need to preserve the metric properties already established via the latent space difference. To aggregate the difference maps along the channel dimension, we found the weighted average proposed by Zhang et al. (2018) to work very well. Thus,

we use one learnable weight to control the importance of a feature. The weight is a multiplier for the corresponding difference map before summation along the channel dimension. To preserve non-negativity and the triangle inequality, the weights are clamped to be non-negative. A negative weight would mean that a larger difference in this feature produces a smaller overall distance, which is not helpful. For spatial and layer aggregation, functions like a summation or averaging are sufficient and generally interchangeable. We tested more intricate aggregation functions, such as a learned spatial average or determining layer importance weights dynamically from the inputs. When the base network is fixed and the metric only has very few trainable weights, this did improve the overall performance. But with a fully trained base network the feature extraction seems to automatically adopt these aspects, making a more complicated aggregation unnecessary.

Additional details for the steps above are given in App. A, where we also show that the proposed Siamese architecture by construction qualifies as a pseudometric.

## 4 DATA GENERATION AND TRAINING

Similarity data sets for natural images typically rely on changing already existing images with distortions, noise or other operations, and assigning ground truth distances according to the strength of the operation. Since we can control the data creation process for numerical simulations directly by altering the simulation, we can generate large amounts of simulation data with growing dissimilarities. In this way, the data contains more information about the nature of the problem, i.e., which changes of the data distribution should lead to increased distances, than by applying modifications as a post-process.

**Data Generation**  Given a set of model equations, e.g. a PDE from fluid dynamics, typical solution methods consist of a solver that, given a set of boundary conditions, computes discrete approximations of the necessary differential operators. The discretized operators and the boundary conditions typically contain problem dependent parameters which we collectively denote with $p_0, p_1, \ldots, p_i, \ldots$ in the following. We only consider time dependent problems, and our solvers start with initial conditions at $t_0$ to compute a series of time steps $t_1, t_2, \ldots$ until a target point in time $(t_t)$ is reached. At that point we obtain a reference output field $o_0$ from one of the PDE variables, e.g., a velocity.

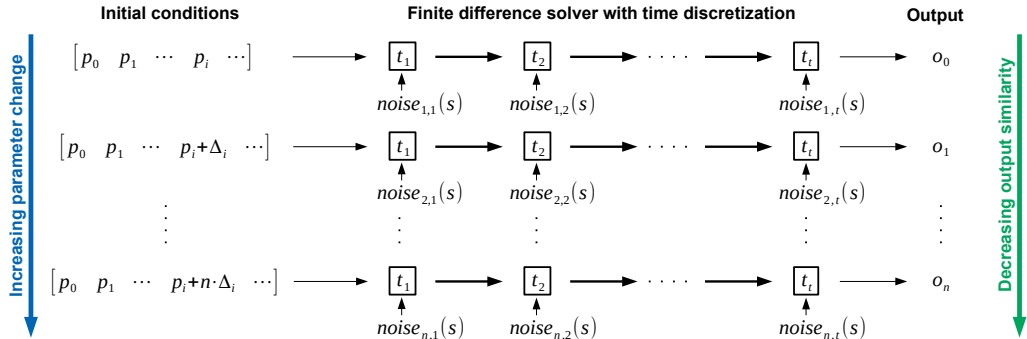

**Figure 3:** General data generation method from a PDE solver for a time dependent problem. With increasing changes of the initial conditions for a parameter $p_i$ in $\Delta_i$ increments, the outputs decrease in similarity. Controlled Gaussian $noise$ is injected in a simulation field of the solver. The difficulty of the learning task can be controlled by scaling $\Delta_i$ as well as the noise strength $s$.

For data set generation, we now incrementally change a single parameter $p_i$ in $n$ steps $\Delta_i, 2 \cdot \Delta_i, \ldots, n \cdot \Delta_i$ to create a series of $n$ outputs $o_1, o_2, \ldots, o_n$. We consider a series obtained this way to be increasingly different from $o_0$. To create natural variations of the resulting data distributions, we add Gaussian noise fields with zero mean and adjustable variance to an appropriate simulation field such as a velocity. This noise allows us to generate a large number of varied data samples for a single simulation parameter $p_i$. In addition, it is similar in nature to numerical errors introduced by discretization schemes. Thus, these perturbations enlarge the space covered by the

training data, and we found that training networks with suitable noise levels improves robustness, as we will demonstrate below. The process for data generation is summarized in Fig. 3.

As PDEs can model extremely complex and chaotic behaviour, there is no guarantee that the outputs always get increasingly dissimilar with the increasing parameter change. This behaviour is what makes the task of similarity assessment so challenging. Even if the solutions are essentially chaotic, their behaviour is not arbitrary but rather governed by the rules of the underlying PDE. For our data set, we choose a range of representative PDEs: We include a pure Advection-Diffusion model (AD) and Burger's equation (BE) which introduces a viscosity term. Furthermore, we use the full Navier-Stokes equations (NSE) which introduce a conservation of mass constraint. When combined with a deterministic solver and a suitable parameter step size, all these PDEs exhibit chaotic behaviour at small scales, so that medium and large scale characteristics of the solutions shift smoothly with increasing changes of the parameters $p_i$. The noise $n$ amplifies the chaotic behaviour to larger scales to create an environment with a controlled amount of perturbations. This lets the network learn about the nature of the chaotic behaviour of PDEs, without overwhelming it with data where patterns are not observable anymore. The latter can easily happen when $\Delta$ or $n$ grow too large and produce essentially random outputs. Instead, we specifically target solutions which are difficult to evaluate in terms of a shallow metric. We choose the smallest $\Delta$ and $n$ such that the ordering of several random output samples with respect to their $L^2$ difference drops below a correlation value of 0.8.

**Training** For training, the scalar 2D fields from the simulations were individually normalized to the $[0, 255]$ range. To produce additional variation, the data is augmented with random color maps, flips, rotations, and a random crop to obtain an input size of $224 \times 224$. Afterwards, each input is normalized to a standard normal distribution by subtracting the mean and dividing by the standard deviation (pre-computed from all available training data). Unless noted otherwise, networks were trained for 40 epochs with an Adam optimizer using a learning rate of $10^{-5}$ that was reduced to $5 \cdot 10^{-6}$ after 15 epochs. To evaluate the networks, only a bilinear resizing and the normalization step is applied to validation or test inputs.

## 5 CORRELATION LOSS FUNCTION

The central goal of our networks is to identify relative differences of input pairs produced via numerical simulations. Thus, instead of employing a loss that forces the network to only infer given labels or distance values, we train our networks to infer the ordering of a given sequence of varying inputs $o_1, \ldots, o_n$. We propose to use the Pearson correlation coefficient (see Pearson, 1920) which yields a value in $[-1, 1]$ that measures the linear relationship between two distributions. A value of 1 implies that a linear equation describes their relationship perfectly. We compute this coefficient for a full series of outputs, such that the network can learn to extract features that arrange this data series in the correct ordering.

We train our networks with minibatches consisting of $n$ outputs, and provide a linearly increasing distance distribution $\boldsymbol{c} \in [0, 1]^n$ representing the parameter change. For a distance prediction $\boldsymbol{d} \in [0, \infty)^n$ of our network for one minibatch, we compute the loss with

$$L(\boldsymbol{c}, \boldsymbol{d}) \ = \ \lambda_1 (\boldsymbol{c} - \boldsymbol{d})^2 \ + \ \lambda_2 (1 - \frac{(\boldsymbol{c} - \bar{\boldsymbol{c}}) \cdot (\boldsymbol{d} - \bar{\boldsymbol{d}})}{\|\boldsymbol{c} - \bar{\boldsymbol{c}}\|_2 \|\boldsymbol{d} - \bar{\boldsymbol{d}}\|_2}) \, . \tag{5}$$

Here, the mean of a minibatch distance vector is denoted by $\bar{c}$ or $\bar{d}$ respectively. The first part of the loss is a regular MSE term, which minimizes the difference between predicted and actual distances. The second part is the Pearson correlation coefficient, which is inverted such that the optimization results in a maximization of the correlation. As this formulation depends on the batch size, the terms in Eq. (5) can be scaled to adjust their relative influence with $\lambda_1$ and $\lambda_2$. For our training, we have ensured a constant batch size due to the fixed number of 10 variations for each reference simulation comprising one batch. If the batch size should be allowed to vary, the influence of both terms needs to be adjusted accordingly. We found that scaling both terms to a similar order of magnitude worked best in our experiments.

In Fig. 4 we investigate how the proposed loss function compares to other commonly used loss formulations for our full network and a pre-trained network similar to Zhang et al. (2018). In addition to our full loss function, we consider a loss function that replaces the Pearson correlation with a simpler cross-correlation $(\boldsymbol{c} \cdot \boldsymbol{d}) / (\|\boldsymbol{c}\|_2 \|\boldsymbol{d}\|_2)$. We also include networks trained with only the MSE or only

the correlation terms for each of the two variants. As shown in Fig. 4, a simple MSE loss yields a low accuracy of less than 0.6. Using any correlation based loss function for the *AlexNet*$_{frozen}$ metric improves the results, but there is no major difference due to the limited number of only 1152 trainable weights. For *LNSM*, the proposed combination of MSE loss with the Pearson correlation performs significantly better than using cross-correlation or a variant without the MSE loss. Interestingly, combining cross correlation with MSE yields worse results than cross correlation alone. This happens because the cross correlation not only affects the ordering but also impacts the absolute distance values. In combination with MSE,

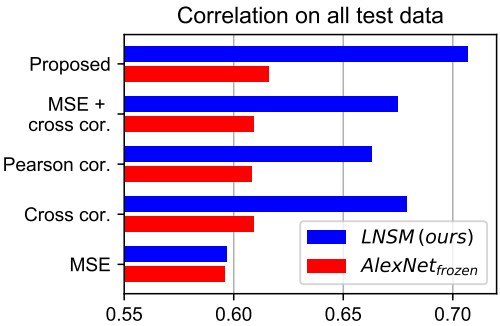

**Figure 4:** Performance on our test data for the proposed approach (*LNSM*) and a smaller model (*AlexNet*$_{frozen}$) using different loss functions.

this can lead to influences that cancel each other. For our loss, the Pearson correlation only handles the ordering while MSE deals with the absolute distances.

## 6 RESULTS

**Data Sets** Using the data generation approach described in Section 4, we created four training (Smo, Liq, Adv and Bur) and two test data sets (LiqN and AdvD) with ten parameter steps for each reference simulation. Based on two 2D NSE solvers, the smoke and liquid simulation training sets (Smo and Liq) add noise to the velocity field and feature varied initial conditions such as fluid position or obstacle properties, in addition to variations of buoyancy and gravity forces. The two other training sets (Adv and Bur) are based on a 1D solvers for AD and BE, concatenated over time to form a 2D result. In both cases, noise was injected into the velocity field, and the varied parameters are changes to the field initialization and forcing functions.

For the test data set, we substantially change the data distribution by injecting noise into the density instead of the velocity field for AD simulations to obtain the AdvD data set, and by including background noise for the velocity field of a liquid simulation (LiqN).

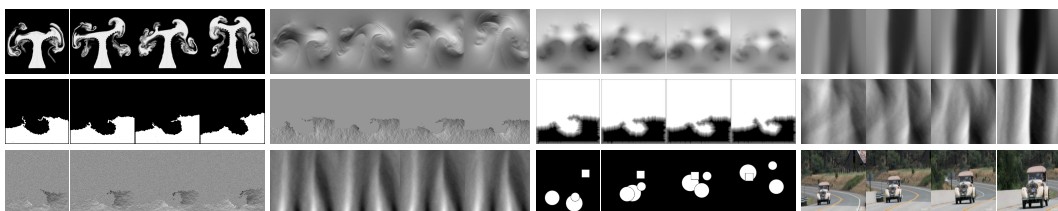

**Figure 5:** Samples from our data sets. For each subset the reference is on the left, and three variations in equal parameter steps follow. From left to right and top to bottom: Smo (density, velocity, and pressure), Adv (density), Liq (flags, velocity, and levelset), Bur (velocity), LiqN (velocity), AdvD (density), Sha and Vid.

In addition, we employed three more test sets (Sha, Vid, and TID) created without PDE models to explore the generalization for data far from our training data setup. We include a shape data set that features multiple randomized moving rigid shapes, a video data set consisting of frames from random video footage, and the perceptual image data set TID2013 from Ponomarenko et al. (2015). All test sets, excluding TID due to a non-comparable structure, are combined into one data set (All) as well. Simulation examples for each data set are shown in Fig. 5 and generation details with additional samples can be found in App. D.

**Performance Evaluation** To evaluate the performance of a metric on a data set, we first compute the distances from each reference simulation to all corresponding variations. Then, the predicted and the ground truth distance distributions over all samples are combined and compared using Spearman's rank correlation coefficient (see Spearman, 1904). Like the Pearson correlation it is a

value in $[-1, 1]$ to compare distributions, but it measures the correlation between ranking variables, i.e. monotonic relationships.

The top part of Tab. 1 shows the performance of the shallow metrics $L^2$ and *SSIM*, as well as the *LPIPS* metric (Zhang et al., 2018) for all our data sets. The results clearly show that shallow metrics are not suitable to compare the samples in our data set, and only achieve good correlation values on the TID2013 data set which contains a large number of pixel-based image variations without contextual structures. The perceptual *LPIPS* metric performs better in general and outperforms our method on the image data sets Vid and TID. This is not surprising, as *LPIPS* is specifically trained for such images. For the simulation data sets LiqN and Sha, however, it performs significantly worse than for the image content. The last row of Tab. 1 shows the results of our *LNSM* network, with a very good performance across all data sets. Note that even though it was not trained with any natural images it still performs well for the image test data sets.

**Table 1:** Performance comparison of existing metrics (top block), experimental designs (middle block), and variants of the proposed method (bottom block) on validation and test data sets measured in terms of Spearman's rank correlation coefficient. **Bold** values show the best performing metric for each data set and ***bold+italic*** values are within a 0.01 error margin of the best performing. On the right a visualization of the combined test data results is shown for selected models.

| Metric | Validation data sets | | | | Test data sets | | | | | | |
|---|---|---|---|---|---|---|---|---|---|---|---|
| | Smo | Liq | Adv | Bur | TID | LiqN | AdvD | Sha | Vid | All | |
| $L^2$ | 0.67 | 0.80 | 0.72 | 0.59 | *0.83* | 0.72 | 0.58 | 0.50 | 0.77 | 0.56 | |
| *SSIM* | 0.67 | 0.75 | *0.75* | 0.68 | 0.80 | 0.25 | **0.70** | 0.35 | 0.70 | 0.48 | |
| *LPIPS v0.1.* | *0.72* | 0.75 | *0.75* | **0.72** | 0.80 | 0.63 | 0.60 | 0.81 | **0.82** | 0.66 | |
| $AlexNet_{random}$ | 0.64 | 0.75 | 0.67 | 0.64 | **0.84** | 0.64 | 0.67 | 0.61 | 0.78 | 0.62 | |
| $AlexNet_{frozen}$ | 0.67 | 0.70 | 0.68 | 0.70 | 0.79 | 0.40 | 0.64 | 0.84 | *0.81* | 0.62 | |
| *Optical flow* | 0.62 | 0.57 | 0.36 | 0.37 | 0.55 | 0.49 | 0.28 | 0.61 | 0.75 | 0.48 | |
| *Non-Siamese* | **0.72** | 0.82 | *0.76* | 0.69 | 0.29 | 0.73 | 0.62 | 0.60 | 0.72 | 0.63 | |
| $Skip_{from\ scratch}$ | 0.65 | **0.84** | 0.74 | 0.67 | 0.80 | 0.79 | 0.59 | 0.83 | 0.79 | *0.70* | |
| $LNSM_{noiseless}$ | 0.64 | *0.83* | 0.74 | 0.60 | 0.80 | **0.81** | 0.58 | 0.84 | 0.76 | 0.69 | |
| $LNSM_{strong\ noise}$ | 0.63 | 0.82 | 0.71 | 0.61 | 0.80 | 0.78 | 0.50 | 0.80 | 0.77 | 0.65 | |
| *LNSM (ours)* | 0.68 | 0.82 | **0.76** | 0.70 | 0.78 | *0.80* | 0.61 | **0.85** | 0.76 | **0.71** | |

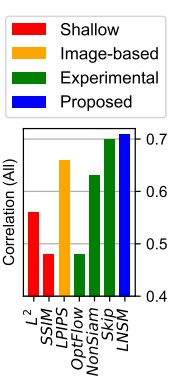

The middle block of Tab. 1 contains several interesting variants: $AlexNet_{random}$ and $AlexNet_{frozen}$ are small models, similar to Zhang et al. (2018), where the base network is the original AlexNet with pre-trained weights. $AlexNet_{random}$ contains purely random aggregation weights without training, while $AlexNet_{frozen}$ only has trained weights are for the channel aggregation. The random model performs surprisingly well, pointing to powers of the underlying CNN architecture, while $AlexNet_{frozen}$ lacks enough trainable weights to fully adjust to the data distribution of the numerical simulations.

Recognizing that many PDEs include transport phenomena, we investigated optical flow (Horn & Schunck, 1981) as means to compute motion from field data. For the *Optical flow* metric we used FlowNet2 (Ilg et al., 2016) to bidirectionally compute the optical flow field between two inputs. The flow fields are aggregate to a single distance value by adding the magnitude of all flow vectors. On the data sets Sha and Vid that are similar to the training data of FlowNet2 it performs relatively well, but on most other data it performs poorly. This shows that computing a simple warping from one input to the other is not enough for a stable metric, although it seems like an intuitive solution. A more robust metric needs the knowledge of the underlying features and their changes to generalize better to new data.

For the *Non-Siamese* metric, we used a non-Siamese architecture that directly predicts the distance from both inputs, to evaluate whether a Siamese architecture is really beneficial. For this purpose, we employed a modified version of AlexNet that reduces the weights of the feature extractor by 50% and of the remaining layers by 90%. As expected, this metric works great on the validation data, but has huge problems with generalization. In addition, even simple metric properties like symmetry are no longer guaranteed because this architecture does not have the inherent constraints of the Siamese setup. Finally, we experimented with multiple fully trained base networks. As re-training existing feature extractors only provided small improvements, we used a custom base network with skip connections for the $Skip_{from\ scratch}$ metric. Its results already come close to the proposed approach on most data sets. App. B contains additional details for the experimental models above.

The last block in Tab. 1 shows variants of the proposed approach, trained with varied noise levels. This inherently changes the difficulty of the data sets. Hence $LNSM_{noiseless}$ was trained with relatively simple data without perturbations, while $LNSM_{strong\,noise}$ was trained with strongly varying data. Both cases decrease the generalizing capabilities of the trained model, resulting in a deteriorated performance for the test data. This indicates that the network needs to see a certain amount of variation at training time in order to become robust, but overly large changes hinder the learning of useful features. We provide a more detailed analysis of varying noise levels in App. C and additional evaluations in App. E.

## 7 CONCLUSION

We have presented the *LNSM* metric to reliably and robustly compare outputs from numerical simulation methods. Our method significantly outperforms existing shallow metric functions and provides better results than other learned metrics on our test data. We demonstrated the usefulness of the correlation loss and the robustness to natural data errors provided via controlled data generation.

Our trained *LNSM* network has the potential to impact a wide range of fields: from the fast and robust accuracy assessment of new simulation methods, over robust optimizations of PDE parameters for real-world data sets, to guiding generative models of physical systems. Furthermore, it will be highly interesting to evaluate other loss functions for learning specialized metrics, e.g. mutual information (Bachman et al., 2019) or contrastive predictive coding (Hénaff et al., 2019). We also plan to evaluate our approach for an even larger collection of PDEs, and to train *LNSM* models for 3D and 4D data sets. Especially unsteady, turbulent flow fields are a highly relevant and interesting area for future work on learned evaluation metrics.

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

# Appendix: Learning Similarity Metrics for Numerical Simulations

This supplemental document contains an analysis of the proposed metric design with respect to properties of metrics in general (App. A), and details to the used network architectures (App. B). Afterwards, material that deals with the data sets is provided. It contains examples and failure cases for each of the data domains and analyzes the impact of the data difficulty (App. C and D). Finally, we explore additional evaluations (App. E) to strengthen the results from Tab. 1, give an overview on the used notation (App. F), and provide a download link for further material (App. G).

## A DISCUSSION OF METRIC PROPERTIES

To analyze if the proposed method qualifies as a *metric*, it is split in two functions $m_1 : \mathbb{I} \to \mathbb{L}$ and $m_2 : \mathbb{L} \times \mathbb{L} \to [0, \infty)$ which operate on the input space $\mathbb{I}$ and the latent space $\mathbb{L}$. Through flattening elements from the input or latent space into vectors, $\mathbb{I} \simeq \mathbb{R}^a$ and $\mathbb{L} \simeq \mathbb{R}^b$ where $a$ and $b$ are the dimensions of the input data or all feature maps respectively, and both values have a similar order of magnitude. $m_1$ describes the non-linear function computed by the base network combined with the following normalization and returns a point in the latent space. $m_2$ uses two points in the latent space to compute a final distance value, so it includes the latent space difference and the aggregation in the spatial, layer, and channel dimensions. With the Siamese network architecture the resulting function for the entire approach is

$$m(\boldsymbol{x}, \boldsymbol{y}) \; = \; m_2(m_1(\boldsymbol{x}), m_1(\boldsymbol{y})).$$

Property (4) mainly depends on $m_1$ because even if $m_2$ itself guarantees this property, $m_1$ could still be non-injective, which means it can map different inputs to the same point in latent space $\tilde{\boldsymbol{x}} = \tilde{\boldsymbol{y}}$ for $\boldsymbol{x} \neq \boldsymbol{y}$. Due to the complicated nature of $m_1$ it is difficult to make accurate predictions about the injectivity of $m_1$. Each base network layer of $m_1$ recursively processes the result of the preceding layer with various feature extracting operations so intuitively, significant changes in the input should produce different feature map results in some layer. But very small changes in the input do lead to zero valued distances predicted by the CNN (i.e. an identical latent space for different inputs), meaning $m_1$ is in practice not injective. In an additional experiment, the proposed architecture was evaluated on about 3500 random inputs from all our data sets, where the CNN received one unchanged and one slightly modified input. The modification consisted of multiple pixel adjustments by one bit (on 8-bit color images) in random positions and channels. When adjusting only a single pixel in the $224 \times 224$ input, the CNN predicts a zero valued distance on about 23% of the inputs, but we never observed an input where seven or more changed pixels resulted in a distance of zero in all experiments.

In this context, the problem of numerical errors is important, because even two slightly different latent space representations could lead to a result that seems to be zero if the difference vanishes in the aggregation operations or is smaller than the floating point precision. On the other hand, an automated analysis to find points that have a different input but an identical latent space image is a challenging problem and left as future work.

The evaluation of the base network and the normalization is deterministic, and hence $\forall \boldsymbol{x}$ : $m_1(\boldsymbol{x}) = m_1(\boldsymbol{x})$ holds. Further, we know that $m(\boldsymbol{x}, \boldsymbol{x}) = 0$ if $m_2$ guarantees that $\forall m_1(\boldsymbol{x})$ : $m_2(m_1(\boldsymbol{x}), m_1(\boldsymbol{x})) = 0$. Thus, the remaining properties (1), (2), and (3) only depend on $m_2$, since for them the original inputs are not relevant, only their respective images in the latent space. The resulting structure with a relaxed identity of indiscernibles is called a *pseudometric*, where $\forall \tilde{\boldsymbol{x}}, \tilde{\boldsymbol{y}}, \tilde{\boldsymbol{z}} \in \mathbb{L}$:

$$m_2(\tilde{\boldsymbol{x}}, \tilde{\boldsymbol{y}}) \; \geq \; 0 \tag{6}$$
$$m_2(\tilde{\boldsymbol{x}}, \tilde{\boldsymbol{y}}) \; = \; m_2(\tilde{\boldsymbol{y}}, \tilde{\boldsymbol{x}}) \tag{7}$$
$$m_2(\tilde{\boldsymbol{x}}, \tilde{\boldsymbol{y}}) \; \leq \; m_2(\tilde{\boldsymbol{x}}, \tilde{\boldsymbol{z}}) + m_2(\tilde{\boldsymbol{z}}, \tilde{\boldsymbol{y}}) \tag{8}$$
$$m_2(\tilde{\boldsymbol{x}}, \tilde{\boldsymbol{x}}) \; = \; 0 \tag{9}$$

Notice, that $m_2$ has to fulfill these properties with respect to the latent space and not the input space. If $m_2$ is carefully constructed the metric properties still apply, independently of the actual design of the base network or the feature map normalization.

A first observation concerning $m_2$ is that if all aggregations were sum operations and the element-wise latent space difference was the absolute value of a difference operation, $m_2$ would be equivalent to computing the $L^1$-norm of the difference vector in latent space.

$$m_2^{sum}(\tilde{\boldsymbol{x}}, \tilde{\boldsymbol{y}}) \;=\; \sum_{i=1}^{b} |\tilde{\boldsymbol{x}}_i - \tilde{\boldsymbol{y}}_i|$$

Similarly, adding a square operation to the elementwise distance in the latent space and computing the square root at the very end leads to the $L^2$-norm of the latent space difference vector. In the same way, it is possible to use any $L^p$-norm with the corresponding operations.

$$m_2^{sum}(\tilde{\boldsymbol{x}}, \tilde{\boldsymbol{y}}) \;=\; \left( \sum_{i=1}^{b} |\tilde{\boldsymbol{x}}_i - \tilde{\boldsymbol{y}}_i|^p \right)^{\frac{1}{p}}$$

In both cases, this forms the metric induced by the corresponding norm which by definition has all desired properties (6), (7), (8), and (9). If we change all aggregation methods to a weighted average operation, each summand is multiplied by a weight $w_i$. This is even possible with learned weights, as they are constant at evaluation time, if they are clamped to be positive as described above. Now, $w_i$ can be attributed to both inputs by distributivity, meaning each input is elementwise multiplied with a constant vector before applying the metric, which leaves the metric properties untouched. The reason is that it is possible to define new vectors in the same space, equal to the scaled inputs. This renaming trivially provides the correct properties.

$$m_2^{weighted}(\tilde{\boldsymbol{x}}, \tilde{\boldsymbol{y}}) \;=\; \sum_{i=1}^{b} w_i |\tilde{\boldsymbol{x}}_i - \tilde{\boldsymbol{y}}_i| \;\overset{w_i \geq 0}{=}\; \sum_{i=1}^{b} |w_i \tilde{\boldsymbol{x}}_i - w_i \tilde{\boldsymbol{y}}_i|$$

Accordingly, doing the same with the $L^p$-norm idea is possible, each $w_i$ just needs a suitable adjustment before distributivity can be applied, keeping the metric properties once again.

$$
\begin{aligned}
m_2^{weighted}(\tilde{\boldsymbol{x}}, \tilde{\boldsymbol{y}}) \;&=\; \left( \sum_{i=1}^{b} w_i |\tilde{\boldsymbol{x}}_i - \tilde{\boldsymbol{y}}_i|^p \right)^{\frac{1}{p}} \\
&=\; \left( \sum_{i=1}^{b} w_i |\tilde{\boldsymbol{x}}_i - \tilde{\boldsymbol{y}}_i| \, |\tilde{\boldsymbol{x}}_i - \tilde{\boldsymbol{y}}_i| \, \ldots |\tilde{\boldsymbol{x}}_i - \tilde{\boldsymbol{y}}_i| \right)^{\frac{1}{p}} \\
&=\; \left( \sum_{i=1}^{b} w_i^{\frac{1}{p}} |\tilde{\boldsymbol{x}}_i - \tilde{\boldsymbol{y}}_i| \, w_i^{\frac{1}{p}} |\tilde{\boldsymbol{x}}_i - \tilde{\boldsymbol{y}}_i| \, \ldots w_i^{\frac{1}{p}} |\tilde{\boldsymbol{x}}_i - \tilde{\boldsymbol{y}}_i| \right)^{\frac{1}{p}} \\
&\overset{w_i \geq 0}{=}\; \left( \sum_{i=1}^{b} |w_i^{\frac{1}{p}} \tilde{\boldsymbol{x}}_i - w_i^{\frac{1}{p}} \tilde{\boldsymbol{y}}_i|^p \right)^{\frac{1}{p}}
\end{aligned}
$$

With these weighted terms for $m_2$, it is possible to describe all used aggregations and latent space difference methods. The proposed method deals with multiple higher order tensors instead of a single vector, so the weights $w_i$ additionally depend on constants like the direction of the aggregations and their position in the latent space tensors. But it is easy to see that mapping a higher order tensor to a vector and keeping track of additional constants still retains all properties in the same way. As a result, the described architecture by design yields a pseudometric that is suitable for comparing simulation data in a way that corresponds to our intuitive understanding of distances.

## B  ARCHITECTURES

The following sections provide details regarding the architecture of the base network and some experimental designs mentioned in Tab. 1.

## B.1  BASE NETWORK DESIGN

Fig. 6 shows the architecture of the base network for the *LNSM* metric. Its purpose is extracting features from both inputs of the Siamese architecture that are useful for the further processing steps. To maximise the usefulness and avoid feature maps that show too similar features, the chosen kernel size and stride of the convolutions is important. Starting with a very larger kernels and strides means the network has a big receptive field and can extract large scale features. For the two following layers, the large strides are replaced by additional MaxPool operations that serve a similar purpose and reduce the spatial size of the feature maps.

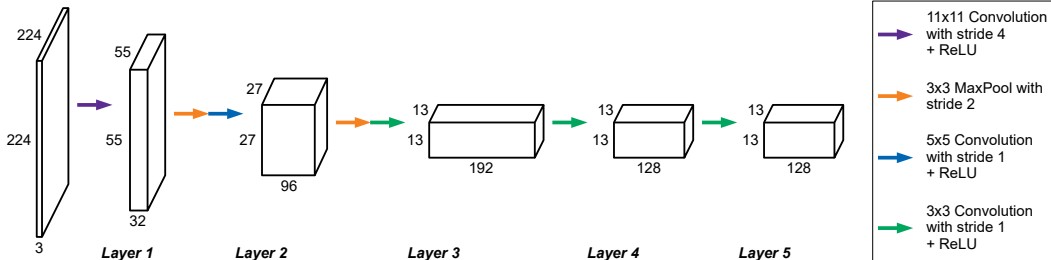

**Figure 6:** Proposed base network architecture consisting of five layers with up to 192 feature maps that are decreasing in spatial size. It is similar to the feature extractor from AlexNet as identical spatial dimensions for the feature maps are used, but it reduces the number of feature maps for each layer by 50% to have fewer weights.

For the three final layers only small convolution kernels and strides are used, but the number of channels is significantly larger than before. The reason is that the very deep features, that typically contain only small scale structures, are most important to distinguish small changes in the inputs. Keeping the number of trainable weights as low as possible was an important consideration for this design, to prevent overfitting to certain simulations types and increase generality. We explored a weight range by using the same architecture and only scaling the number of feature maps in each layer. The final design shown in Fig. 6 with about 0.62 million weights worked best.

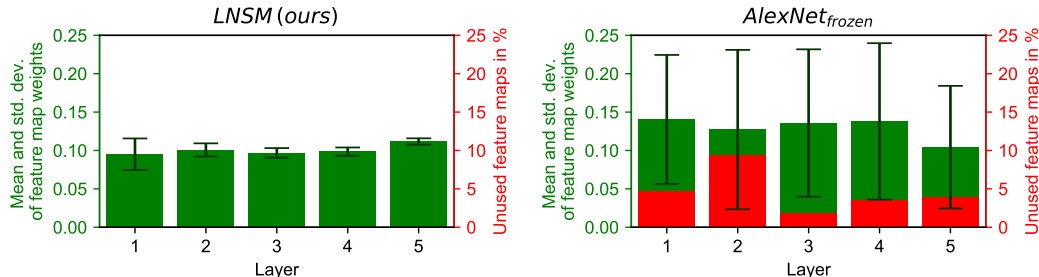

**Figure 7:** Analysis of the distributions of learned feature map aggregation weights across the base network layers. Displayed is our method for fully training the base network (left) in comparison to using pre-trained weights (right).

In the following, we analyze the contributions of the per-layer features of two different metric networks to highlight differences in terms of how the features are utilized for the distance estimation task. In Fig. 7 the learned feature map aggregation weights of our *LNSM* network show a very similar mean and a small standard deviation throughout the five layers. This means, all feature maps similarly contribute to establishing the distances, and the aggregation just fine-tunes the relative importance of each feature. In addition, all features receive a weight greater than zero, and as a result no feature is excluded from contributing to the final distance value.

Employing a fixed pre-trained feature extractor on the other hand shows a very different picture: Although the mean across the different network layers is similar, the contributions of different features vary strongly, which is visible in the standard deviation being significantly larger. Furthermore, 2–10% of the feature maps in each layer receive a weight of zero and hence were deemed not useful

at all for establishing the distances. This illustrates the usefulness of a targeted network in which all features contribute to the distance inference.

## B.2 FEATURE MAP NORMALIZATION

Here we analyze how feature maps are best normalized for the "normalization" step of our algorithm, i.e., the red ellipses in Fig. 2. Assume we have a $3^{\text{rd}}$ order feature tensor $\mathbf{G}$ with dimensions $(g_c, g_x, g_y)$ from one layer of the base network (see Fig. 2). We can form a series $\mathbf{G}_0, \mathbf{G}_1, \dots$ for all possible forms of this tensor in our training samples (computed as a preprocessing step).

Below, we evaluate three different normalization methods and also consider not normalizing at all, denoted by

$$norm_{none}(\mathbf{G}) = \mathbf{G}.$$

The normalization only happens in the channel dimension, so all following operations accumulate values along $(:, g_x, g_y)$ while keeping $g_x$ and $g_y$ constant, i.e., are applied independently of the spatial dimensions. The unit length normalization proposed by Zhang et al. (2018)

$$norm_{unit}(\mathbf{G}) \;=\; \frac{\mathbf{G}}{\|\mathbf{G}\|_2}$$

only considers the current sample. In this case $\|\mathbf{G}\|_2$ is a $2^{\text{nd}}$ order tensor with the Euclidean norms of $\mathbf{G}$ along the channel dimension. Com-

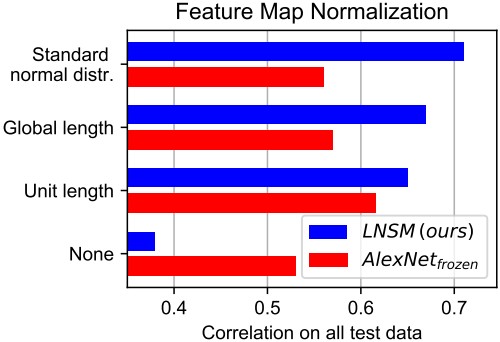

**Figure 8:** Performance on our test data for the proposed approach (*LNSM*) and a smaller model (*AlexNet$_{frozen}$*) using different normalizations.

bined with summation as the aggregation operation in channel direction, this results in a cosine distance which only measures angles of the latent space vectors. Using a learned average instead means the angles are no longer uniform, but warped according to the importance of each feature (i.e. the resulting angle changes differently for the same amount of change in two separate features). Extending this idea to consider other training samples as well, leads to a global unit length normalization

$$norm_{global}(\mathbf{G}) \;=\; \frac{\mathbf{G}}{\max\left(\|\mathbf{G}_0\|_2, \|\mathbf{G}_1\|_2, \dots\right)}$$

where the maximum Euclidean norm of all available samples is employed. As a result, not only the angle of the latent space vectors, but also their magnitude compared to the largest feature vector is available in the aggregation. This formulation is not really robust yet, because the largest feature vector could be an outlier w.r.t. the typical content. Instead, we can consider the full feature vector as a normal distribution and transform it to a standard normal distribution with the proposed

$$norm_{dist.}(\mathbf{G}) \;=\; \frac{\mathbf{G} - \text{mean}\left(\|\mathbf{G}_0\|_2, \|\mathbf{G}_1\|_2, \dots\right)}{\text{std}\left(\|\mathbf{G}_0\|_2, \|\mathbf{G}_1\|_2, \dots\right)}.$$

In addition to the angle, this formulation allows for a robust comparison of the magnitude of each feature vector in the global magnitude distribution. Fig. 8 shows a comparison of these normalization methods on the combined test data. Using no normalization is detrimental in both cases as succeeding operations can not reliably compare the features. Interestingly, the unit length normalization works best for the *AlexNet$_{frozen}$* metric (similar to *LPIPS* from Zhang et al. (2018)) that only uses learned aggregation weights with a fixed AlexNet feature extractor. This observation allows for a conclusion about the features extracted by AlexNet. For the original task of image classification, the magnitude of a feature vector does not seem to carry information about the feature. Interpreting the length as part of the feature for our task in $norm_{global}$ and $norm_{dist.}$, obviously harms the performance of the metric. Therefore, training the feature extractor such that the magnitude of the feature vectors bears some meaning should improve the results for the complex normalizations. The performance of our approach with a fully trained feature extractor in Fig. 8 shows exactly this behaviour: A more complex normalization directly yields better results since the features can be adapted to utilize it.

### B.3 Recursive "Meta-Metric"

Since comparing the feature maps, i.e., the yellow ellipse in Fig. 2, is a central operation of the proposed metric calculations, we experimented with replacing it with an existing CNN-based metric. In theory, this would allow for a recursive, arbitrarily deep network that repeatedly invokes itself:

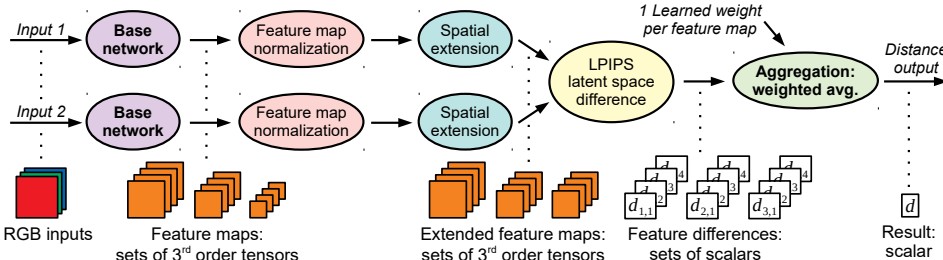

**Figure 9:** Adjusted distance computation for a *LPIPS*-based latent space difference. To provide sufficiently large inputs for *LPIPS*, small feature maps are spatially enlarged with nearest neighbor interpolation. In addition, *LPIPS* creates scalar instead of spatial differences leading to a simplified aggregation.

first, the deep representations of inputs are used, then the deep representations of the deep representations, etc. In practice, however, using more than one recursion step is currently not feasible due to increasing computational requirements in addition to vanishing gradients.

Fig. 9 shows how our computation method can be modified for a CNN-based latent space difference, instead of an elementwise operation. Here we employ *LPIPS* (Zhang et al., 2018). There are two main differences compared to Fig. 2: First, the *LPIPS* latent space difference creates single distance values for a pair of feature maps instead of a spatial feature difference. As a result, the following aggregation is a single learned average operation and spatial or layer aggregations are no longer necessary. We also performed experiments with a spatial *LPIPS* version here, but due to memory limitations these were not successful. Second, the convolution operations in *LPIPS* have a lower limit for spatial resolution, and some feature maps of our base network are quite small (see Fig. 6). Hence, we up-scale the feature maps below the required spatial size of $32 \times 32$ using nearest neighbor interpolation.

On our combined test data, such a metric with a fully trained base network results in a correlation value of 0.62 (compare to Tab. 1). Overall, it achieves a performance comparable to *AlexNet$_{random}$* or *AlexNet$_{frozen}$*.

### B.4 Optical Flow Metric

In the following, we describe our approach to compute a metric via optical flow (OF). For an efficient OF evaluation we employed a pre-trained network (Ilg et al., 2016). From an OF network $f : \mathbb{I} \times \mathbb{I} \to \mathbb{R}^{i_{max} \times j_{max} \times 2}$ with two inputs data fields $\boldsymbol{x}, \boldsymbol{y} \in I$, we get the flow vector field $f^{\boldsymbol{xy}}(i,j) = (f_1^{\boldsymbol{xy}}(i,j), f_2^{\boldsymbol{xy}}(i,j))^T$, where $i$ and $j$ denote the location and $f_1$ and $f_2$ denote the components of the flow vectors. In addition, we have a second flow field $f^{\boldsymbol{yx}}(i,j)$ computed from the reversed input ordering. We can now define a function $m : \mathbb{I} \times \mathbb{I} \to [0, \infty)$:

$$m(\boldsymbol{x}, \boldsymbol{y}) = \sum_{i=0}^{i_{max}} \sum_{j=0}^{j_{max}} \sqrt{(f_1^{\boldsymbol{xy}}(i,j))^2 + (f_2^{\boldsymbol{xy}}(i,j))^2} + \sqrt{(f_1^{\boldsymbol{yx}}(i,j))^2 + (f_2^{\boldsymbol{yx}}(i,j))^2}$$

Intuitively, this function computes the sum over the magnitudes of all flow vectors in both vector fields. With this definition, it is obvious that $m(\boldsymbol{x}, \boldsymbol{y})$ fulfills the metric properties of non-negativity and symmetry (see Eq. (1) and (2)). Under the assumption that identical inputs create a zero flow field, a relaxed identity of indiscernibles holds as well (see Eq. (9)). Compared to the proposed approach there is no guarantee for the triangle inequality though, so $m(\boldsymbol{x}, \boldsymbol{y})$ only qualifies as a pseudo-semimetric.

Fig. 10 shows flow visualizations on data examples produced by FlowNet2. The metric works relatively well for inputs that are similar to the training data from FlowNet2, like the shape data

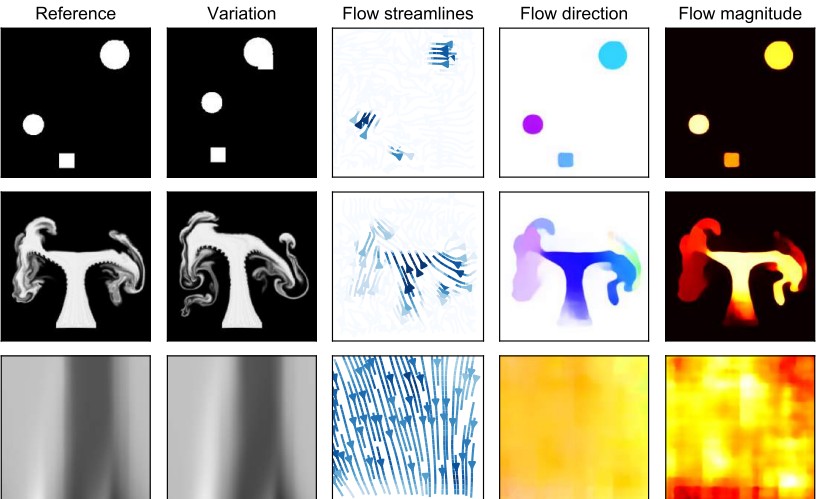

**Figure 10:** Outputs from FlowNet2 on data examples. The flow streamlines are sparse visualization of the resulting flow field and indicate the direction of the flow by their orientation and its magnitude by their color (darker being larger). The two visualizations on the right show the dense flow field and are colorcoded to show the flow direction (blue/yellow: vertical, green/red: horizontal) and the flow magnitude (brighter being larger).

example in the top row. For data that provides some outline, for instance the smoke simulation example in the middle row or also liquid data, the metric does not work as well but still provides a reasonable flow field. But for full spatial examples like from the Burger's or Advection-Diffusion equation (see bottom row) the network is no longer able to produce meaningful flow fields. The results are often a very uniform flow with similar magnitude and direction. These observations are reflected in the performance of the *Optical flow* metric in Tab. 1.

### B.5 NON-SIAMESE ARCHITECTURE

To compute a metric without the Siamese architecture outlined above, we use a network structure with a single output, as shown in Fig. 11. Thus, instead of having two identically feature extractors and combining the feature maps, here the distance is directly predicted from the stacked inputs with a single network with about 1.24 million weights. After using the same feature extractor as described in Section B.1, the final set of feature maps is spatially reduced with an adaptive MaxPool operation. Next, the result is flattened and three consecutive fully connected layers process the data to form the final prediction. Here, the last activation function is a sigmoid instead of ReLU. The reason is that a ReLU would clamp every negative intermediate value to a zero distance, while a sigmoid compresses the intermediate value to a small distance that is more meaningful than directly clamping it.

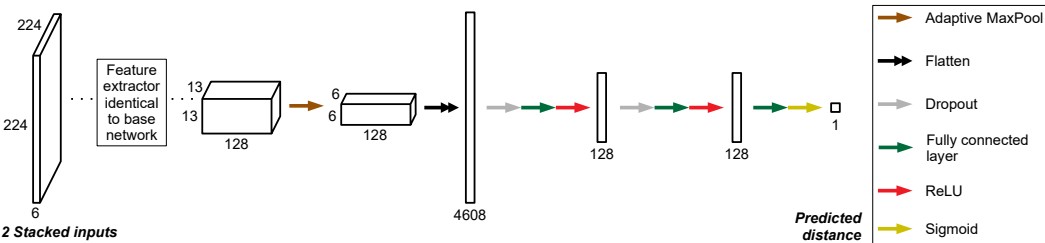

**Figure 11:** Non-Siamese network architecture with the same feature extractor used in Fig. 6. It uses both stacked inputs and directly predicts the final distance value from the last set of feature maps with several fully connected layers.

In terms of metric properties, this architecture only provides non-negativity (see Eq. (1)) due to the final sigmoid function. All other properties can not be guaranteed without further constraints. This is the main disadvantage of a non-Siamese network. These issues could be alleviated with specialized training data or by manually adding constraints to the model, e.g., to have some amount of symmetry (see Eq. (2)) and at least a weakened identity of indiscernibles (see Eq. (9)). However, compared to a Siamese network that guarantees them by design, these extensions are clearly sub-optimal. As a result of the missing properties, this network has significant problems with generalization. While it performs well on the training data, the performance noticeably deteriorates for several of the test data sets, as shown in Tab. 1.

### B.6 Skip Connections in Base Network

As explained above, our base network primarily serves as a feature extractor to produce activations that are employed to evaluate a learned metric. In many state-of-the-art methods, networks with

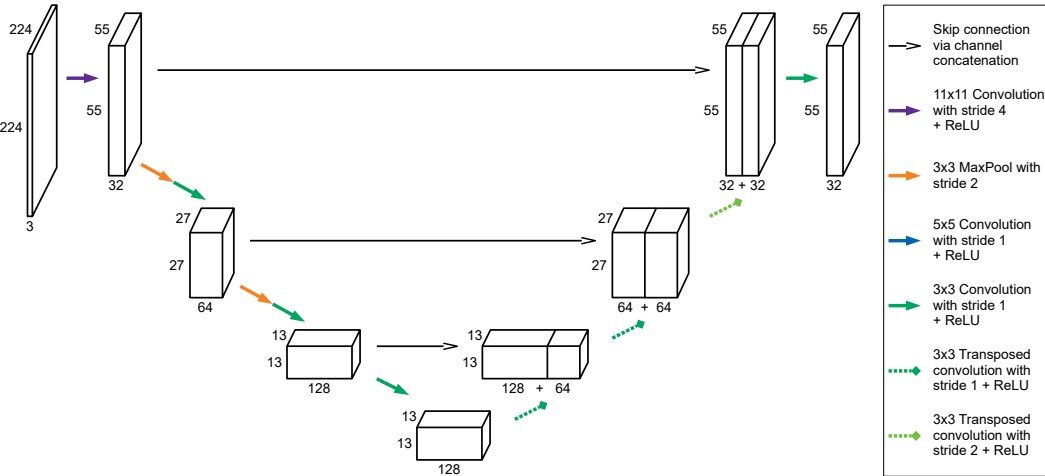

**Figure 12:** Network architecture with skip connections for better information transport between feature maps. Transposed convolutions are used to upscale the feature maps in the second half of the network to match the spatial size of earlier layers for the skip connections.

skip connections are employed (Ronneberger et al., 2015; He et al., 2016; Huang et al., 2017), as experiments have shown that these connections help to preserve information from the inputs. In our case, the classification "output" of a network such as the AlexNet plays no actual role. Rather, the features extracted along the way are crucial. Hence, skip connections should not improve the inference task for our metrics. To verify that this is the case, we have included tests with a base network similar to the popular UNet architecture (Ronneberger et al., 2015). For our experiments, we kept the early layers closely in line with the feature extractors that worked well for the base network (see Section B.1). Only the layers in the decoder part have an increased spatial feature map size to accommodate the skip connections. As expected, this network can be used to compute reliable metrics for the input data without negatively affecting the performance. However, as expected, the improvements of skip connections for regular inference tasks do not translate into improvements for the metric calculations, as shown in Tab. 1.

## C Impact of Data Difficulty

We shed more light on the aspect of noise levels and data difficulty via six reduced data sets, that consist of a smaller amount of Smoke and Advection-Diffusion data with differently scaled noise strength values. Results are shown in Fig. 13. Increasing the noise level creates more difficult data as shown by the dotted and dashed plots representing the performance of the $L^2$ and the *LPIPS* metric on each data set. Both roughly follow an exponentially decreasing function. Each point on the solid line plot is the test result of a reduced *LNSM* model trained on the data set with the corresponding

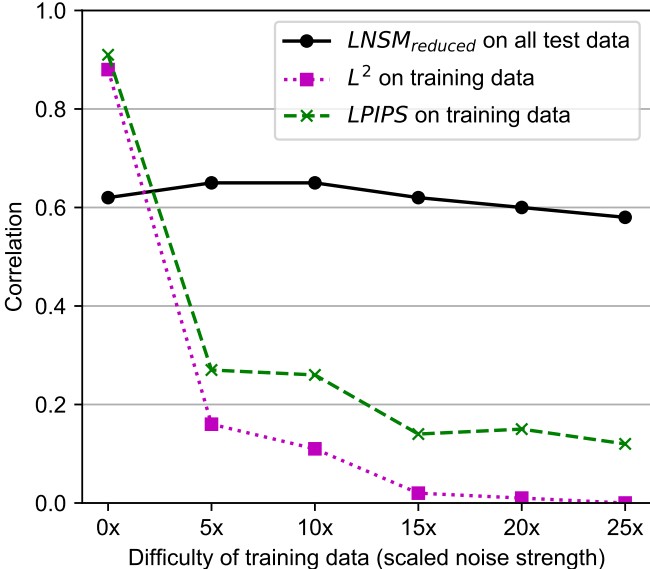

**Figure 13:** Impact of increasing data difficulty for a reduced training data set. Evaluations on training data for $L^2$ and *LPIPS*, and the test performance of models trained with the different reduced data sets (*LNSM_{reduced}*) are shown.

noise level. Apart from the data, the entire training setup was identical. This shows that the training process is very robust to the noise, as the result on the test data only slowly decreases for very high noise levels. Furthermore, small amounts of noise improve the generalization compared to a the model that was trained without any noise. This is somewhat expected, as a model that never saw noisy data during training can not learn to extract features which are robust w.r.t. noise.

## D    DATA SET DETAILS

In the following sections the generation of each used data set is described. For each figure showing data samples (consisting of a reference simulation and several variants with a single changing initial parameter), the leftmost image is the reference and the images to the right show the variants in order of increasing parameter change. For the figures 14, 15, 16, and 17 the first subfigure (a) demonstrates that medium and large scale characteristics behave very non-chaotic for simulations without any added noise. They are only included for illustrative purposes and are not used for training. The second and third subfigure (b) and (c) in each case show the training data of *LNSM*, where the large majority of data falls into the category (b) of normal samples that follow the generation ordering, even with more varying behaviour. Category (c) is a small fraction of the training data and the shown examples are specifically picked, worst case examples to show how the chaotic behaviour can sometimes override the ordering intended by the data generation. In some cases, category (d) is included to show how normal data samples from the test set differ from the training data.

### D.1    NAVIER-STOKES EQUATIONS

These equations describe the general behaviour of fluids with respect to advection, viscosity, pressure, and mass conservation. Eq. (10) defines the conservation of momentum and Eq. (11) the conservation of mass inside the fluid.

$$\frac{\partial u}{\partial t} + (u \cdot \nabla)u = -\frac{\nabla P}{\rho} + \nu \nabla^2 u + g \tag{10}$$

$$\nabla \cdot u = 0 \tag{11}$$

In this context, $u$ is the velocity, $P$ is the pressure the fluid exerts, $\rho$ is the density of the fluid (usually assumed to be constant), $\nu$ is the kinematic viscosity coefficient that indicates the thickness of the fluid, and $g$ denotes the acceleration due to gravity. With this PDE three data sets were created using

a smoke and a liquid solver. For all data, 2D simulations were run until a certain step and useful data fields were exported afterwards.

**Smoke** For the smoke data, a standard Eulerian fluid solver using a preconditioned pressure solver based on conjugate gradient and a Semi-Lagrangian advection scheme was employed. The gen-

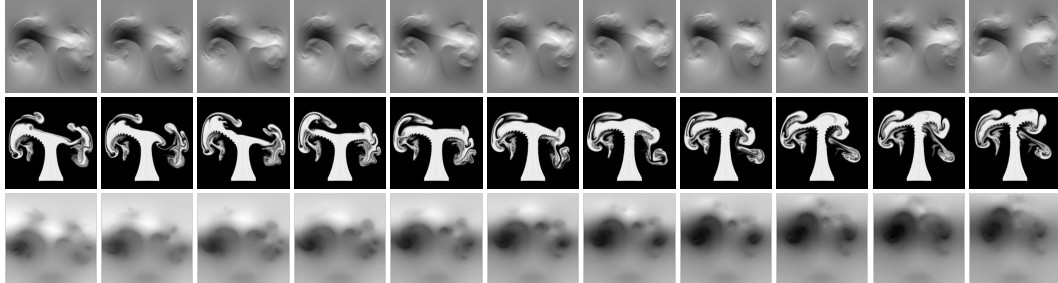

**(a)** Data samples generated without noise: tiny output changes following generation ordering

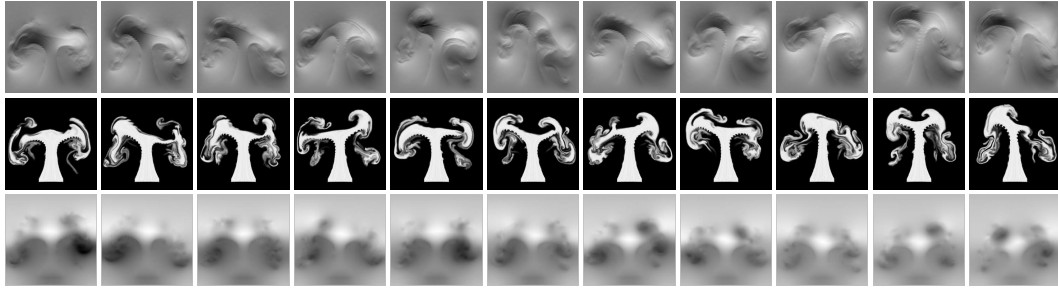

**(b)** Normal training data samples with noise: larger output changes but ordering still applies

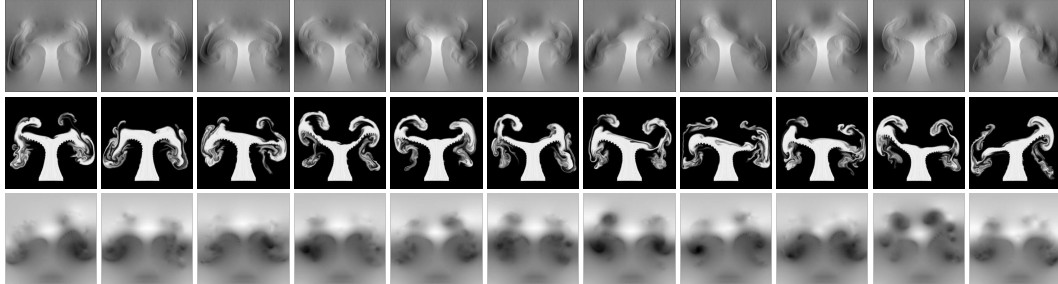

**(c)** Outlier data samples: noise can override the generation ordering by chance

**Figure 14:** Various smoke simulation examples using one component of the velocity (top rows), the density (middle rows), and the pressure field (bottom rows).

eral setup for every smoke simulation consists of a rectangular smoke source at the bottom with a fixed additive noise pattern to provide smoke plumes with more details. Additionally, there is a downwards directed, spherical force field area above the source which divides the smoke in two major streams along it. We chose this solution over an actual obstacle in the simulation, in order to avoid overfitting to a clearly defined black obstacle area inside the smoke data. Once the simulation reaches a predefined time step, the density, pressure, and velocity field (separated by dimension) is exported and stored. Some examples can be found in Fig. 14. With this setup the following initial conditions were varied in isolation:

- Smoke buoyancy in x- and y-direction
- Strength of noise added to the velocity field
- Amount of force in x- and y-direction provided by the force field
- Orientation and size of the force field
- Position of the force field in x- and y-direction
- Position of the smoke source in x- and y-direction

**Liquid** For the liquid data, a solver based on the fluid implicit particle (FLIP) method proposed by Zhu & Bridson (2005) was employed. It is a Eulerian-Lagrangian hybrid approach that replaces the Semi-Lagrangian advection scheme with particle based advection to achieve higher accuracy and prevent the loss of mass. Still, this method is not optimal as we experienced problems with mass loss, especially for larger noise values.

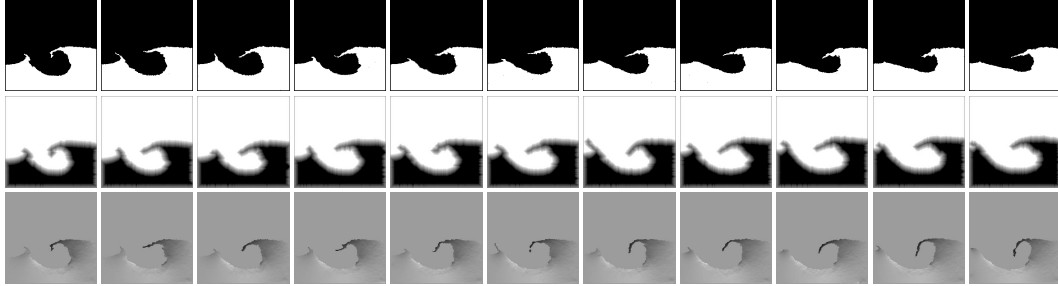

**(a)** Data samples generated without noise: tiny output changes following generation ordering

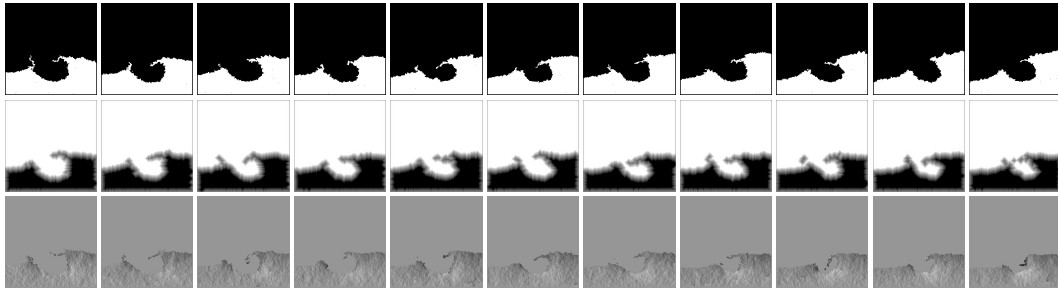

**(b)** Normal training data samples with noise: larger output changes but ordering still applies

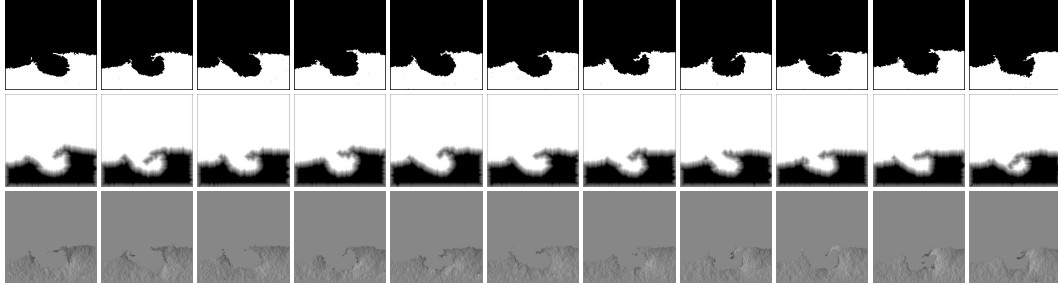

**(c)** Outlier data samples: noise can override the generation ordering by chance

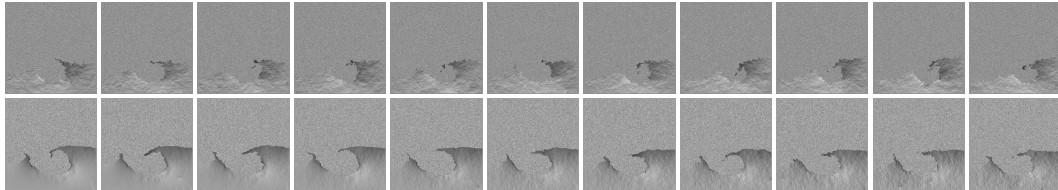

**(d)** Data samples from test set with additional background noise

**Figure 15:** Several liquid simulation examples using the binary indicator flags (top rows), the extrapolated levelset values (middle rows), and one component of the velocity field (bottom rows) for the training data and only the velocity field for the test data.

The simulation setup consists of a large breaking dam and several smaller liquid areas for more detailed splashes. After the dam hits the simulation boundary a large, single drop of liquid is created in the middle of the domain that hits the already moving liquid surface. Then, the extrapolated level set values, binary indicator flags, and the velocity field (separated by dimension) are saved, with some examples shown in Fig. 15. The list of varied parameters include:

- Radius of the liquid drop
- Position of the drop in x- and y-direction
- Amount of additional gravity force in x- and y-direction
- Strength of noise added to the velocity field

For the liquid test set additional background noise was added to the velocity field of the simulations (see Fig. 15d). Because this only alters the velocity field, the extrapolated level set values and binary indicator flags are not used for this data set.

### D.2 ADVECTION-DIFFUSION AND BURGER'S EQUATION

For these PDEs our solvers only discretized and solve the corresponding equation in 1D. Afterwards, the different time steps of the solution process are concatenated along a new dimension to form 2D data with one spatial and one time dimension.

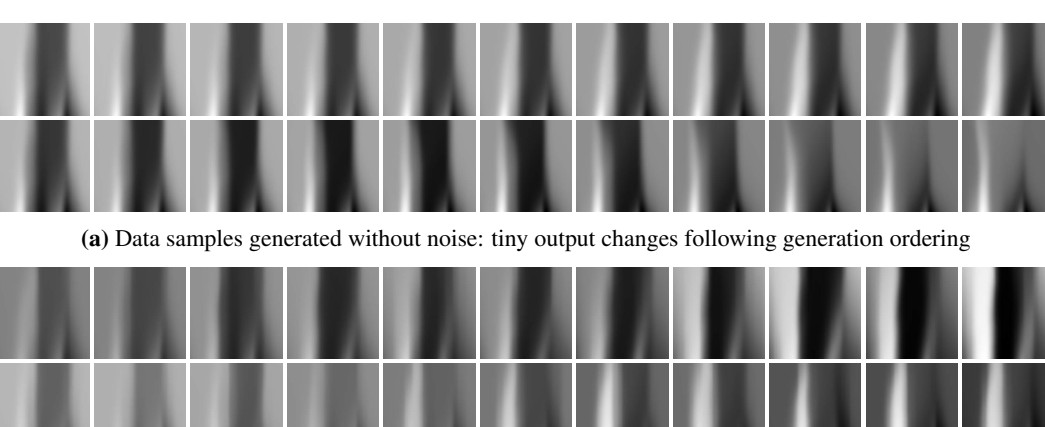

**(a)** Data samples generated without noise: tiny output changes following generation ordering

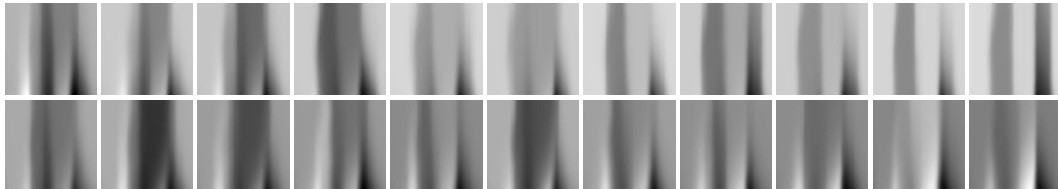

**(b)** Normal training data samples with noise: larger output changes but ordering still applies

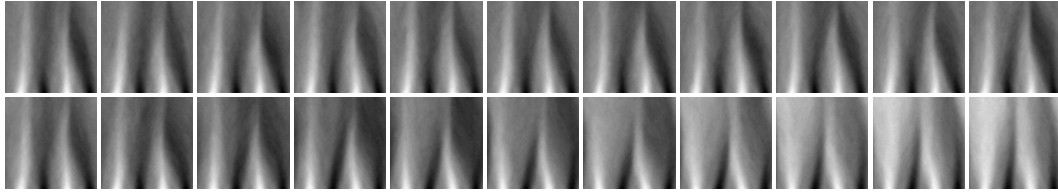

**(c)** Outlier data samples: noise can override the generation ordering by chance

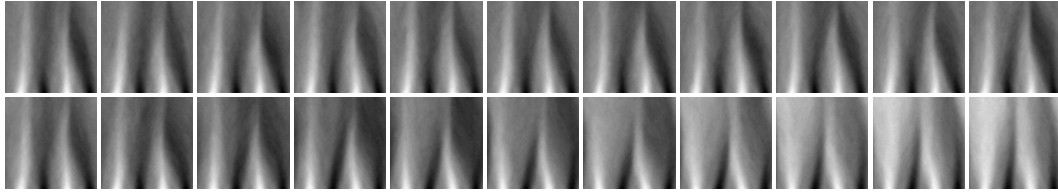

**(d)** Data samples from test set noise directly added to the density field

**Figure 16:** Various examples from the Advection-Diffusion equation using the density field.

**Advection-Diffusion Equation**    This equation describes how a passive quantity is transported inside a velocity field due to the processes of advection and diffusion. Eq. (12) is the simplified Advection-Diffusion equation with constant diffusivity and no sources or sinks.

$$\frac{\partial d}{\partial t} = \nu \nabla^2 d - u \cdot \nabla d \tag{12}$$

Here, $d$ denotes the density, $u$ is the velocity, and $\nu$ is the kinematic viscosity (also known as diffusion coefficient) that determines the strength of the diffusion. Our solver employed a simple implicit

time integration and a diffusion solver based on conjugate gradient without preconditioning. The initialization for the 1D fields of the simulations was created by overlaying multiple parameterized sine curves with random frequencies and magnitudes.

In addition, continuous forcing controlled by further parameterized sine curves was included in the simulations over time. In this case, the only initial conditions to vary are the forcing and initialization parameters of the sine curves and the strength of the added noise. From this PDE only the passive density field was used as shown in Fig. 16. For the Advection-Diffusion test set the noise was instead added directly to the passive density field of the simulations. This creates results with more small scale details as shown in Fig. 16d.

**Burger's Equation** This equation is very similar to the Advection-Diffusion equation and describes how the velocity field itself changes due to diffusion and advection.

$$\frac{\partial u}{\partial t} = \nu \nabla^2 u - u \cdot \nabla u \tag{13}$$

Eq. (13) is known as the viscous form of the Burger's equation that can develop shock waves, and again $u$ is the velocity and $\nu$ denotes the kinematic viscosity. Our solver for this PDE used a slightly different implicit time integration scheme, but the same diffusion solver as used for the Advection-Diffusion equation. The simulation setup and parameters were also the same; the only difference is

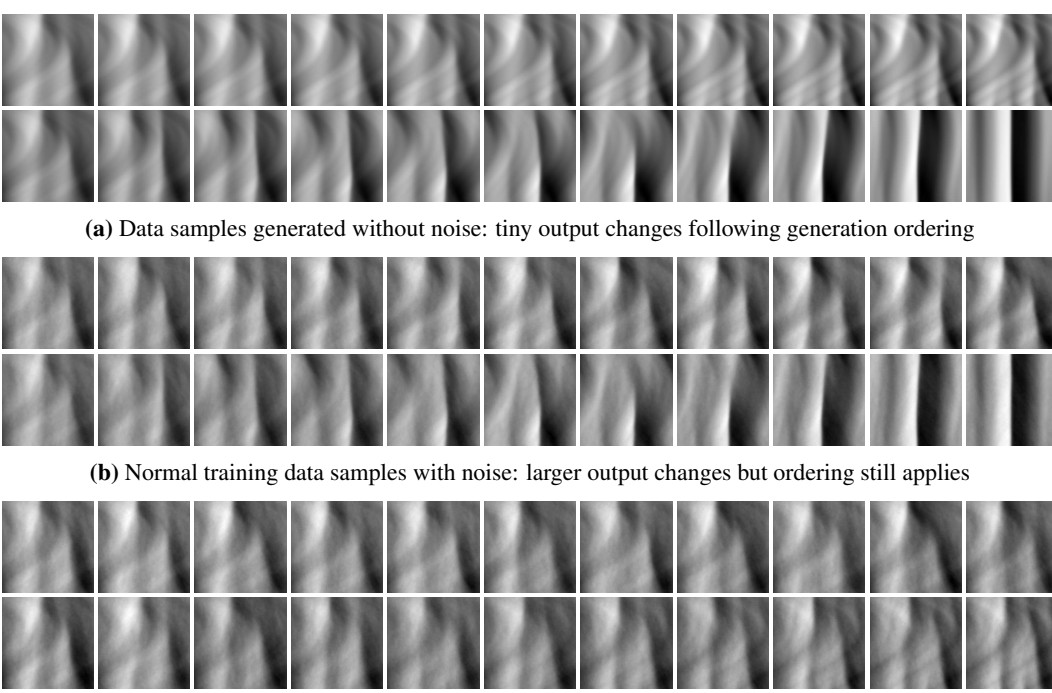

**(a)** Data samples generated without noise: tiny output changes following generation ordering

**(b)** Normal training data samples with noise: larger output changes but ordering still applies

**(c)** Outlier data samples: noise can override the generation ordering by chance

**Figure 17:** Different simulation examples from the Burger's equation using the velocity field.

that the velocity field instead of the density is exported. As a consequence, the data in Fig. 17 looks relatively similar to the results from the Advection-Diffusion equation.

### D.3 OTHER

The remaining data sets are not based on PDEs and thus not generated with the proposed method. The data is only used to test the generalization of the discussed metrics and not for training or validation.

**Shapes** This data set tests if the metrics are able to track simple, moving geometric shapes. To create it, a straight path between two random points inside the domain is generated and a random shape is moved along this path in steps of equal distance. The size of the used shape depends on the

distance between start and end point, such that a significant fraction of the shape overlaps between two consecutive steps. It is also ensured that no part of the shape leaves the domain at any step, by using a sufficiently big boundary area when generating the path.

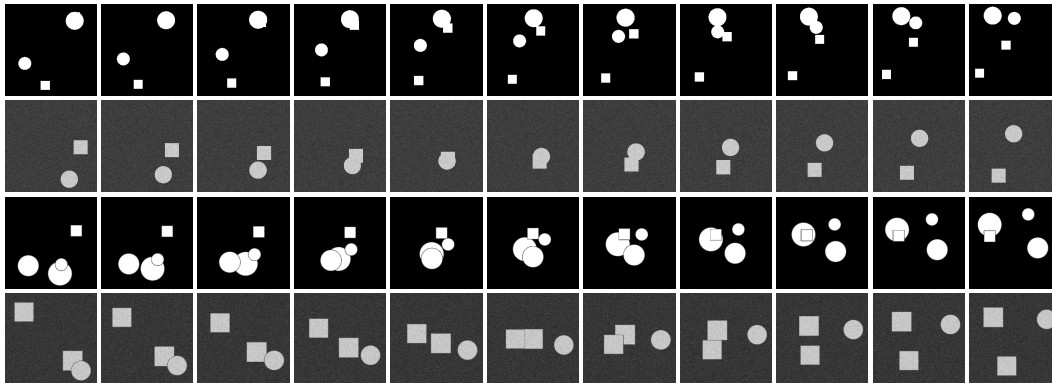

**Figure 18:** Examples from the shapes data set using a field with only binary shape values (first row), shape values with additional noise (second row), smoothed shape values (third row), and smoothed values with additional noise (fourth row).

With this method, multiple random shapes for a single data sample are produced and their paths can overlap, such that they occlude each other to provide an additional challenge. All shapes are moved in their parametric representation and only when exporting the data, they are discretized onto a fixed binary grid. To add more variations to this simple approach, we also apply them in a non-binary way with smoothed edges and include additive gaussian noise over the entire domain. Examples for the different exports can be seen in Fig. 18.

**Video**   For this data set, different publicly available video recordings were acquired and processed in three steps. First, videos with abrupt cuts, scene transitions or camera movements were discarded,

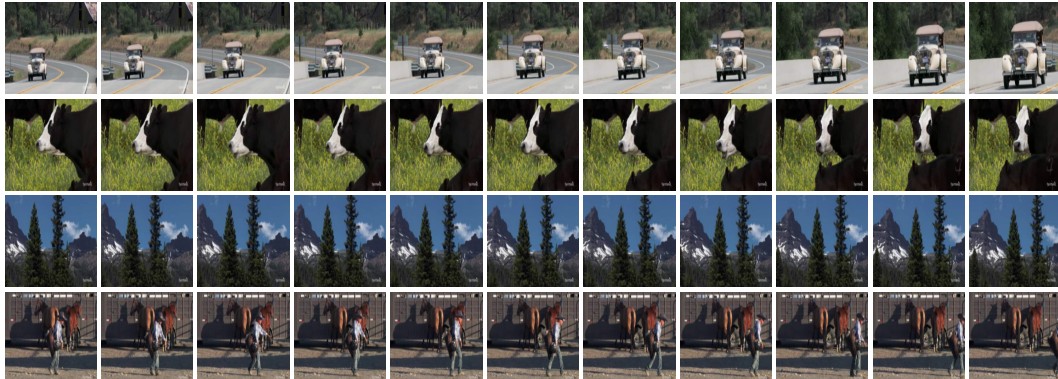

**Figure 19:** Multiple examples from the video data set.

and afterwards the footage was broken down into single frames. Then, each frame was resized to match the spatial size of our other data by linear interpolation. Since directly using consecutive frames is no challenge for any analyzed metric and all of them recovered the ordering almost perfectly, we achieved a more meaningful data set by skipping several intermediate frames. For the final data set, we defined the first frame of every video as the reference and subsequent frames in an interval step of ten frames as the increasingly different variations. Some data examples can be found in Fig. 19.

**TID2013**   This data set was created by Ponomarenko et al. (2015) and used without any further modifications. It consists of 25 reference images with 24 distortion types in five levels. As a result it is not directly comparable to our data sets, so it is excluded from the test set aggregations in Tab. 1.

The distortions focus on various types of noise, image compression and color changes. Fig. 20 contains examples from the data set.

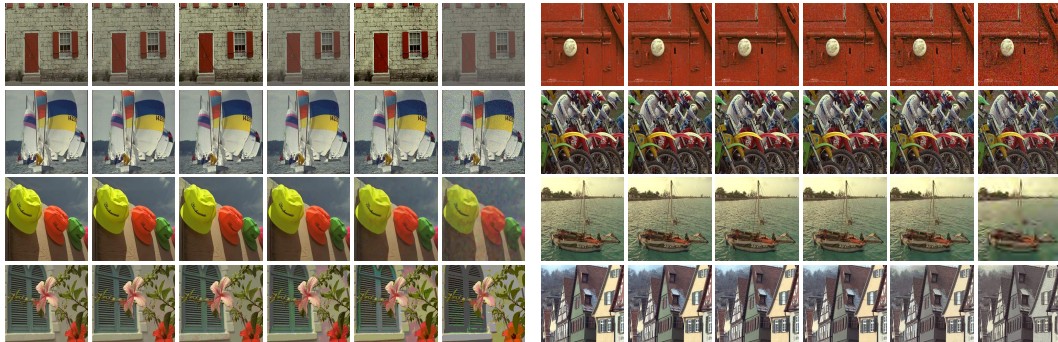

**Figure 20:** Examples from the TID2013 data set proposed by Ponomarenko et al. (2015). Displayed are a change of contrast, three types of noise, denoising, jpg2000 compression, and two color quantizations (from left to right and top to bottom).

## D.4 DATA AUGMENTATION

To add more variation to our data, we include different data augmentation techniques. These augmentations help the network to become more invariant to typical data transformations. Examples for the augmented data can be found in Fig. 21. In order to compare to existing natural image feature extractors trained for RGB images, we focus on three channel inputs. As a first step, the scalar, i.e. grey-scale, data is converted to a three channel RGB input with a random color map (out of five fixed variations) or no color map at all by copying the data directly to each channel.

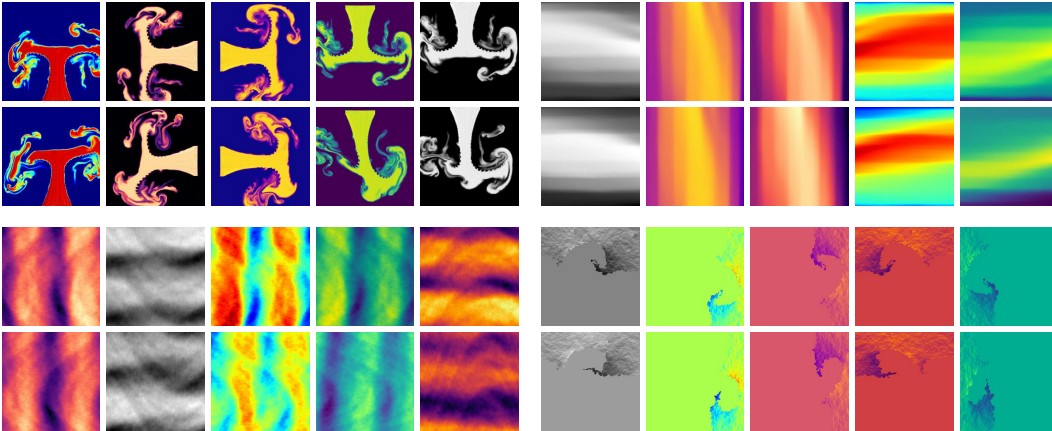

**Figure 21:** Augmented samples from the training sets in groups with Smoke, Advection-Diffusion Equation, Burger's Equation, and Liquid data. The upper row in each group shows the same reference simulation, and the lower row contains variations with different ground truth distances (increasing from left to right).

Next, the data is randomly flipped on either axis and rotated in increments of $90°$ to provide robustness to rotations. The rotated data fields are then cropped from their simulation size $256 \times 256$ to a size of $224 \times 224$ which is the typical input size for existing feature extractors. Finally, each input is normalized to a standard normal distribution. The mean and standard deviation are computed from all available training data without augmentations in a pre-processing step with an online algorithm from Welford (1962). Note that each data sample gets a new augmentation every time it is used, and that the corresponding reference receives the identical transformations the keep comparability (see Fig. 21). For the validation and test data, only a bilinear interpolation to the correct input size and the final normalization is performed.

# E   ADDITIONAL EVALUATIONS

In the following, we demonstrate other ways to compare the performance of the analyzed metrics on our data sets. In Tab. 2 the Pearson correlation coefficient is used instead of Spearman's rank correlation coefficient. While Spearman's correlation measures monotonic relationships by using ranking variables, it directly measures linear relationships as discussed in Section 5.

The results in Tab. 2 match very closely to the numbers provided in Tab. 1. The best performing metrics in both tables are identical, only the numbers slightly vary. Since a linear and a monotonic relation describes the results of the metrics similarly well, there are no apparent non-linear dependencies that can not be captured using the Pearson correlation.

**Table 2:** Performance comparison on validation and test data sets measured in terms of the Pearson correlation coefficient. **Bold** values show the best performing metric for each data set and ***bold+italic*** values are within a 0.01 error margin of the best performing. On the right a visualization of the combined test data results is shown for selected models.

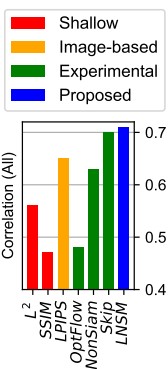

| Metric | Validation data sets | | | | Test data sets | | | | | |
|---|---|---|---|---|---|---|---|---|---|---|
| | Smo | Liq | Adv | Bur | TID | LiqN | AdvD | Sha | Vid | All |
| $L^2$ | 0.66 | 0.81 | 0.71 | 0.58 | **0.85** | 0.72 | 0.57 | 0.55 | 0.76 | 0.56 |
| SSIM | 0.67 | 0.74 | 0.75 | 0.68 | 0.83 | 0.25 | **0.69** | 0.34 | 0.66 | 0.47 |
| LPIPS v0.1. | ***0.71*** | 0.75 | ***0.76*** | **0.72** | 0.79 | 0.63 | 0.61 | 0.82 | **0.80** | 0.65 |
| $AlexNet_{random}$ | 0.63 | 0.75 | 0.66 | 0.64 | 0.80 | 0.63 | 0.65 | 0.68 | 0.77 | 0.60 |
| $AlexNet_{frozen}$ | 0.67 | 0.70 | 0.68 | 0.70 | 0.79 | 0.40 | 0.63 | 0.84 | ***0.80*** | 0.61 |
| Optical flow | 0.63 | 0.56 | 0.37 | 0.39 | 0.49 | 0.45 | 0.28 | 0.61 | 0.74 | 0.48 |
| Non-Siamese | **0.71** | ***0.82*** | 0.75 | 0.69 | 0.26 | 0.72 | 0.62 | 0.65 | 0.68 | 0.63 |
| $Skip_{from\ scratch}$ | 0.65 | **0.83** | 0.74 | 0.66 | 0.72 | 0.78 | 0.59 | 0.83 | 0.78 | ***0.70*** |
| $LNSM_{noiseless}$ | 0.64 | ***0.82*** | 0.74 | 0.60 | 0.69 | **0.80** | 0.58 | 0.83 | 0.75 | 0.68 |
| $LNSM_{strong\ noise}$ | 0.63 | 0.81 | 0.71 | 0.61 | 0.70 | 0.78 | 0.50 | 0.80 | 0.76 | 0.64 |
| LNSM (ours) | 0.68 | ***0.82*** | **0.76** | 0.70 | 0.71 | ***0.79*** | 0.61 | **0.86** | 0.76 | **0.71** |

In the Tables 3 and 4 we employ a different, more intuitive approach to determine combined correlation values for each data set using the Pearson correlation. We no longer analyzing the entire predicted distance distribution and the ground truth distribution at once as done for Tab. 1. Instead,

**Table 3:** Performance comparison on test data sets measured by computing mean and std. dev. (in brackets) of Pearson correlation coefficients from individual data samples. **Bold** values show the best performing metric for each data set and ***bold+italic*** values are within a 0.01 error margin of the best performing.

| Metric | Test data sets | | | | | |
|---|---|---|---|---|---|---|
| | TID | LiqN | AdvD | Sha | Vid | All |
| $L^2$ | ***0.98 (0.19)*** | 0.76 (0.29) | 0.59 (0.45) | 0.61 (0.29) | 0.82 (0.33) | 0.68 (0.37) |
| SSIM | 0.92 (0.40) | 0.26 (0.49) | **0.73 (0.36)** | 0.45 (0.45) | 0.77 (0.37) | 0.57 (0.46) |
| LPIPS v0.1. | 0.94 (0.33) | 0.66 (0.38) | 0.65 (0.41) | 0.83 (0.24) | ***0.85 (0.30)*** | 0.74 (0.36) |
| $AlexNet_{random}$ | 0.96 (0.27) | 0.68 (0.34) | 0.69 (0.38) | 0.68 (0.26) | 0.82 (0.33) | 0.71 (0.34) |
| $AlexNet_{frozen}$ | 0.94 (0.33) | 0.41 (0.49) | 0.67 (0.39) | **0.85 (0.21)** | **0.85 (0.29)** | 0.70 (0.40) |
| Optical flow | 0.74 (0.67) | 0.50 (0.34) | 0.32 (0.53) | 0.63 (0.45) | 0.78 (0.45) | 0.53 (0.49) |
| Non-Siamese | 0.47 (0.88) | 0.76 (0.24) | 0.66 (0.41) | 0.67 (0.28) | 0.76 (0.42) | 0.71 (0.35) |
| $Skip_{from\ scratch}$ | **0.99 (0.14)** | **0.85 (0.15)** | 0.61 (0.42) | 0.84 (0.23) | 0.82 (0.33) | ***0.76 (0.33)*** |
| $LNSM_{noiseless}$ | ***0.98 (0.19)*** | **0.86 (0.15)** | 0.61 (0.41) | 0.84 (0.26) | 0.79 (0.38) | 0.76 (0.34) |
| $LNSM_{strong\ noise}$ | ***0.99 (0.14)*** | 0.83 (0.19) | 0.52 (0.45) | 0.81 (0.23) | 0.82 (0.35) | 0.73 (0.36) |
| LNSM (ours) | 0.97 (0.23) | 0.83 (0.22) | 0.64 (0.42) | **0.86 (0.23)** | 0.80 (0.37) | **0.77 (0.34)** |

we individually compute the correlation between the ground truth and the predicted distances for the single data samples of the data set. From the single correlation values, we compute the mean and standard deviations shown in the tables. Note that this approach potentially produces less accurate

comparison results, as small errors in the individual computations can accumulate to larger deviations in mean and standard deviation. Still, the values in both tables lead to very similar conclusions as Tab. 1: The best performing metrics are almost the same and low combined correlation values match with results that have a high standard deviation and a low mean.

**Table 4:** Performance comparison on validation data sets measured by computing mean and standard deviation (in brackets) of Pearson correlation coefficients from individual data samples. **Bold** values show the best performing metric for each data set and ***bold+italic*** values are within a 0.01 error margin of the best performing. On the right a visualization of the combined test data results is shown for selected models.

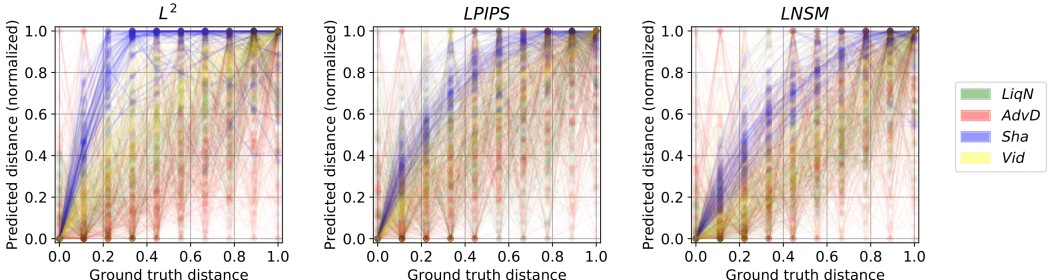

| Metric | Validation data sets | | | |
|---|---|---|---|---|
| | Smo | Liq | Adv | Bur |
| $L^2$ | 0.71 (0.34) | 0.84 (0.23) | 0.76 (0.28) | 0.63 (0.41) |
| *SSIM* | 0.73 (0.30) | 0.78 (0.29) | ***0.80 (0.26)*** | 0.72 (0.38) |
| *LPIPS v0.1.* | **0.77 (0.28)** | 0.79 (0.24) | ***0.81 (0.26)*** | **0.77 (0.32)** |
| *AlexNet$_{random}$* | 0.68 (0.36) | 0.79 (0.28) | 0.71 (0.36) | 0.69 (0.36) |
| *AlexNet$_{frozen}$* | 0.72 (0.31) | 0.74 (0.29) | 0.73 (0.35) | 0.75 (0.33) |
| *Optical flow* | 0.66 (0.38) | 0.59 (0.47) | 0.38 (0.52) | 0.41 (0.49) |
| *Non-Siamese* | ***0.76 (0.27)*** | ***0.87 (0.19)*** | ***0.80 (0.24)*** | 0.75 (0.33) |
| *Skip$_{from scratch}$* | 0.69 (0.34) | **0.87 (0.19)** | 0.79 (0.26) | 0.72 (0.34) |
| *LNSM$_{noiseless}$* | 0.68 (0.33) | 0.85 (0.24) | 0.78 (0.30) | 0.66 (0.37) |
| *LNSM$_{strong noise}$* | 0.67 (0.36) | 0.85 (0.22) | 0.76 (0.33) | 0.67 (0.39) |
| *LNSM (ours)* | 0.72 (0.31) | 0.85 (0.22) | **0.81 (0.23)** | 0.75 (0.33) |

Fig. 22 shows a visualization of predicted distances $c$ against ground truth distances $d$ for different metrics on every sample from the test sets. Each plot contains over 6700 individual data points to illustrate the global distance distributions created by the metrics, without focusing on single cases. A theoretical optimal metric would recover a perfectly narrow distribution along the line $c = d$, while worse metrics recover broader, more curved distributions. Overall, the sample distribution of an $L^2$ metric is very wide. *LPIPS* manages to follow the optimal diagonal a lot better, but our approach approximates it with the smallest deviations, as also shown in the tables above. The $L^2$ metric performs very poorly on the shape data indicated by the too steeply increasing blue lines that flatten after a ground truth distance of 0.3. *LPIPS* already significantly reduces this problem, but *LNSM* still works slightly better. A similar issue is visible for the Advection-Diffusion data, where

**Figure 22:** Distribution evaluation of ground truth distances against normalized predicted distances for $L^2$, *LPIPS* and *LNSM* on all test data (color coded).

for $L^2$ a larger number of red samples is below the optimal $c = d$ line, than for the other metrics. *LPIPS* has the worst overall performance for liquid test set, indicated by the large number of fairly chaotic green lines in the plot. On the video data, all three metrics perform similarly well.

A fine-grained distance evaluation in 200 steps of $L^2$ and our *LNSM* metric via the mean and standard deviation of different data samples is shown in Fig. 23. Similar to Fig. 22, the mean of an optimal metric would follow the ground truth line with a standard deviation of zero, while the mean

of worse metrics deviates around the line with a high standard deviation. The plot on the left combines eight samples with different seeds from the Sha data set, where only a single shape is used. Similarly, the center plot aggregates eight samples from Sha with more than one shape. The right plot shows six data samples from the LiqN test set that vary by the amount of noise that was injected into the simulation (see Fig. 3). The task of only tracking a single shape in the example on the left

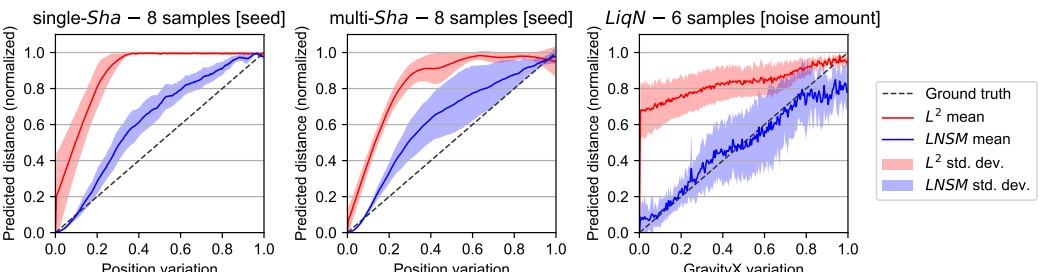

**Figure 23:** Mean and standard deviation of normalized distances over multiple data samples for $L^2$ and *LNSM*. The samples differ by the quantity displayed in brackets. Each data sample uses 200 parameter variation steps instead of 10 like the others in our data sets. For the shape data the position of the shape varies and for the liquid data the gravity in x-direction is adjusted.

is the easiest of the three shown cases. Both metrics have no problem to recover the position change until a variation of 0.4, where $L^2$ can no longer distinguish between the different samples. Our metric recovers distances with a continuously rising mean and a very low standard deviation. The task in the middle is already harder, as multiple shapes can occlude each other during the position changes. Starting at a position variation of 0.4 both metrics have a quite high standard deviation, but the proposed method stays closer to the ground truth line. $L^2$ shows a similar issue as before, because it flattens relatively fast. The plot on the right features the hardest task. Here, both metrics perform similar as each has a different problem in addition to an unstable mean. Our metric stays close to the ground truth, but has a quite high standard deviation starting at about a variation of 0.4. The standard deviation of $L^2$ is lower, but instead it starts off with a big jump from the first few data points. To some degree this is caused by the normalization of the plots, but it still overestimates the relative distances for small variations in the simulation parameter.

These findings also match with the distance distribution evaluations in Fig. 22 and the tables above: Our method has a significant advantage over shallow metrics on shape data, while the differences of both metrics become much smaller for the liquid test set.

## F  NOTATION

In this work, we follow the notation suggested by Goodfellow et al. (2016). Vector quantities are displayed in bold and tensors use a sans-serif font. Double-barred letters indicate sets or vector spaces. The following symbols are used:

| | |
|---|---|
| $\mathbb{R}$ | Real numbers |
| $i, j$ | Indexing in different contexts |
| $\mathbb{I}$ | Input space of the metric, i.e., color images / field data of size $224 \times 224 \times 3$ |
| $a$ | Dimension of the input space $\mathbb{I}$ when flattened to a single vector |
| $\boldsymbol{x}, \boldsymbol{y}, \boldsymbol{z}$ | Elements in the input space $\mathbb{I}$ |
| $\mathbb{L}$ | Latent space of the metric, i.e., sets of $3^{\text{rd}}$ order feature map tensors |
| $b$ | Dimension of the latent space $\mathbb{L}$ when flattened to a single vector |
| $\tilde{\boldsymbol{x}}, \tilde{\boldsymbol{y}}, \tilde{\boldsymbol{z}}$ | Elements in the latent space $\mathbb{L}$, corresponding to $\boldsymbol{x}, \boldsymbol{y}, \boldsymbol{z}$ |
| $w$ | Weights for the learned average aggregation (1 per feature map) |

| | |
|---|---|
| $p_0, p_1, \ldots$ | Initial conditions / parameters of a numerical simulation |
| $n$ | Number of steps when varying a simulation parameter, thus size of a minibatch |
| $o_0, o_1, \ldots, o_n$ | Series of outputs of a simulation with increasing ground truth distance to $o_0$ |
| $\Delta$ | Amount of change in a single simulation parameter |
| $t_1, t_2, \ldots, t_t$ | Time steps of a numerical simulation |
| $s$ | Strength of the noise added to a simulation |
| $\boldsymbol{c}$ | Ground truth distance distribution, determined by the data generation via $\Delta$ |
| $\boldsymbol{d}$ | Predicted distance distribution (supposed to match the corresponding $\boldsymbol{c}$) |
| $\bar{\boldsymbol{c}}, \bar{\boldsymbol{d}}$ | Mean of the distributions $\boldsymbol{c}$ and $\boldsymbol{d}$ |
| $\|\ldots\|_2$ | Euclidean norm of a vector |
| $m(\boldsymbol{x}, \boldsymbol{y})$ | Entire function computed by our metric |
| $m_1(\boldsymbol{x}, \boldsymbol{y})$ | First part of $m(\boldsymbol{x}, \boldsymbol{y})$, i.e., the base network and the feature map normalization |
| $m_2(\tilde{\boldsymbol{x}}, \tilde{\boldsymbol{y}})$ | Second part of $m(\boldsymbol{x}, \boldsymbol{y})$, i.e., the latent space difference and the aggregations |
| $\mathsf{G}$ | $3^{\text{rd}}$ order feature tensor from one layer of the base network |
| $g_c, g_x, g_y$ | Channel dimension ($g_c$) and spatial dimensions ($g_x, g_y$) of $\mathsf{G}$ |
| $f$ | Optical flow network |
| $f^{\boldsymbol{xy}}, f^{\boldsymbol{yx}}$ | Flow fields computed by an optical flow network $f$ from two inputs in $\mathbb{I}$ |
| $f_1^{\boldsymbol{xy}}, f_2^{\boldsymbol{xy}}$ | Components of the flow field $f^{\boldsymbol{xy}}$ |
| $\nabla, \nabla^2$ | Gradient ($\nabla$) and Laplace operator ($\nabla^2$) |
| $\partial$ | Partial derivative operator |
| $t$ | Time in our PDEs |
| $u$ | Velocity in our PDEs |
| $\nu$ | Kinematic viscosity / diffusion coefficient in our PDEs |
| $d, \rho$ | Density in our PDEs |
| $P$ | Pressure in the Navier-Stokes Equations |
| $g$ | Gravity in the Navier-Stokes Equations |

## G  MATERIAL DOWNLOAD

Anonymized and time-stamped supplemental material for our submission can be downloaded **here**.

