# OpenReview forum: "Learning Similarity Metrics for Numerical Simulations"
_ICLR.cc/2020/Conference — Reject_

### Official Review · AnonReviewer2 · 2019-10-23
**Official Blind Review #2**

**Rating:** 3

**Review:**

The paper proposes to use siamese convolutional network to generate a metric for images. The authors use both PDE generated simulated data and some real-world set to evaluate the proposed metric.

While it seems to be a legit metric I have some concerns listed below:

1. Need a proof why the cnn-based evaluation metric is a metric. In particular, the triangle inequality.

2. Instead of using traditional metrics, I can first do a feature transform and then try a metric on top of the transformed feature space. Intuitively I do not quite see how the proposed metric is better than this naive feature+distance design.

3. The paper spend one and a half page describing the data generation process. The authors claim using PDEs to generate data can have some special control of the data de-similarity, on the other hand, the performance on the simulated data does not indicate how practical the designed “metric” is for real-world data which is possibly the main interest of most of the audience.

4. The evaluation on the real-world data set is very limited. How does the proposed metric perform on image datasets such as CIFAR, ImageNet, MNIST? For example we may design experiment comparing inter/intra-class metric comparisons. And how that metric can be used to improve the sota results on the data sets?

5. The metric construction section is not quite self-contained. For example, could the authors states in details how the feature map normalization and aggregations are actually done in the algorithm instead of just citing some related works?

6. The loss function used in the training seems weird. I am worried how those two terms balance. Also it is a batch-dependent loss given the \bar{c} and \bar{d}. As the batch size gets changed the estimation accuracy of the Pearson coefficient may change in a different way as your first squared loss.

7. Pearson coefficient only captures the linear correlation. I would suggest looking into something like mutual information instead.


**Experience Assessment:**

I have published one or two papers in this area.

**Review Assessment: Checking Correctness Of Derivations And Theory:**

I assessed the sensibility of the derivations and theory.

**Review Assessment: Checking Correctness Of Experiments:**

I assessed the sensibility of the experiments.

**Review Assessment: Thoroughness In Paper Reading:**

I read the paper at least twice and used my best judgement in assessing the paper.

---

> ### Author Response · Authors · 2019-11-11
> **Response to Review 2**
>
> We thank reviewer 2 for the review and constructive feedback. Below are our responses to your concerns.
>
> 1. Proof of metric properties
> Yes, this is an important point. A discussion of why the metric satisfies pseudo-metric axioms is included in Appendix A of our original submission.
>
> 2. Advantage over feature transform with a metric in feature space
> Using a feature transformation (via a base network) and a metric in the feature space (via the latent space difference and the aggregations) is what we employ for our LNSM metric. Our paper targets learning specialized features for the content of PDE solutions, which follow a substantially different data distributions than other commonly used data sets, such as natural images.
> Our experiments show that using existing image feature extractors like AlexNet or VGG leads to suboptimal assessments of similarity, and that substantially better results can be achieved with our approach (e.g., see results table in Section 6 and further experiments in Appendix B.1).
>
> 3. Performance of the metric for real-world data
> We agree that demonstrating performance on real-world data is crucial. In our case, the simulated data represents the real world as our approach targets this data domain. To give one specific example: measuring the similarity of different turbulent flow fields from simulations or experiments is a fundamental challenge for research in fluid dynamics. Our data set contains a large quantity of NS solutions, and we believe our approach makes important steps towards establishing neural networks as a tool in this area. At the same time, our evaluation demonstrates that the LNSM metric yields excellent performance for a wide range of real-world advection diffusion problems.
>
> 4. Evaluation on image datasets (CIFAR, ImageNet, MNIST)
> As we are not aiming to create a metric for natural images, but instead for simulation data from PDEs (which typically fundamentally differ from image content), we do not consider image datasets to be real world data. The image based test data we evaluate and discuss in our submission is only included to illustrate that our metric generalizes well to data very far outside the training range. This is important, e.g., for generalization to other types of physical behavior and the corresponding PDE models. Additionally, CIFAR, ImageNet, or MNIST do not provide ground truth “distances” which are essential for our training process and performance evaluations. Instead, we show results for the TID and Vid datasets (in Tables 1 to 3) as representatives of natural images with ground truth distances.
>
> 5. Implementation details for feature map normalization and aggregations
> The feature map normalization is discussed in detail in Appendix B.2. The aggregations along the different dimensions are illustrated in the overview in Figure 2. We are also planning to provide the full source code and data sets soon.
>
> 6. Balance for loss function terms
> This is a valid point: the formulation of our training loss as given in the paper currently relies on a fixed batch size. For inference, this is not an issue, and for training, we have ensured a constant batch size due to the fixed number of variations for each reference simulation comprising a mini-batch. For training, we scaled both terms to be of similar magnitude. If a metric should be trained with different batch sizes, the loss function would need to be adjusted accordingly. We will add this discussion to our document.
>
> 7. Mutual information instead of pearson correlation
> Using mutual information (MI) in the loss function is an interesting idea. As we are primarily interested in an ordering with respect to the strength of the parameter variation, going beyond linear correlation did not seem necessary. Additionally, our approach already outperforms existing metrics for the data domain we are targeting, i.e., PDE solutions. As such, we believe MI-based losses will be an interesting topic for future work.

---

### Official Review · AnonReviewer3 · 2019-10-24
**Official Blind Review #3**

**Rating:** 8

**Review:**

This is a well-written paper which looks into options for learning similarity metrics on data derived from PDE models common in the sciences. In comparison to other metric learning settings, here a type of ground-truth distance information is available (rather than, say, triplets), and it is possible to attempt to directly target an objective function which aims to match the learned distance to the ground-truth distance. The model architecture follows a fairly standard siamese-network setup.

Quite a bit of space is devoted to ensuring that the learned metric actually satisfies pseudo-metric axioms. This is all very clearly presented, with justifications for different modeling choices and how they preserve the axioms; my only criticism here is that many aspects of this are fairly obvious (i.e. an architecture which shares weights in computing the embeddings of both data points, followed by computing a squared L2 distance, will quite clearly get us in the ballpark of a pseudometric), but in my opinion "excess" clarity is much better than the opposite.

I think the more important contribution of the paper is in sections 4 and 5, which outlines a specific data generation process, including means of injecting noise, and compares options for loss functions. I would have expected pearson correlation to work quite well in this context, and it is interesting to note that performance notably improved by also adding an MSE term. I am curious about the "distance" prediction, as described just before equation 5, where it is stated that d \in R^n — is this really R^n? The target distance c is in [0,1]^n, and it seems like a simple modification to the distance prediction network would be capable of ensuring that the predicted values d also fall in this range. Such normalization could reduce the need for the MSE loss term, which presumably helps keep the overall relative scales of the two distances in check.

The empirical testing is also thorough, and I particularly appreciate the use of the random-weight networks as a baseline — I think it is good to note that these are actually fairly competitive on many of the test data sets (in fact, I believe it should be in bold for "TID" in table 1).

I think the main weakness of this paper is that it falls slightly short of actually presenting the real use cases and needs for a similarity metric on PDE outputs — in my opinion, this comes to play when matching the output of a PDE-based model with real data. It would be nice to see a discussion of how this could be useful for parameter inference in PDE models. If there are other important applications of a distance learned in this way, I think the paper could benefit *greatly* by pointing them out. Otherwise, this risks being perceived as adding little value, since for individual PDE runs with known parameters, there is a ground-truth distance available — in which case, why bother using deep learning to estimate the distance, if the parameters are known? I think relevance to applications should to be clearly addressed.

The supplemental material is long, but complete and clearly presented.

**Experience Assessment:**

I have read many papers in this area.

**Review Assessment: Checking Correctness Of Derivations And Theory:**

I assessed the sensibility of the derivations and theory.

**Review Assessment: Checking Correctness Of Experiments:**

I assessed the sensibility of the experiments.

**Review Assessment: Thoroughness In Paper Reading:**

I read the paper at least twice and used my best judgement in assessing the paper.

---

> ### Author Response · Authors · 2019-11-11
> **Response to Review 3**
>
> We thank reviewer 3 for the detailed review and encouraging comments.
>
> Detailed answers:
>
> 1. Distance prediction in R^n
> You are right that the distance predictions are by construction non-negative, so it actually is: d \in [0, infinity)^n. We deliberately chose not to use an upper bound to the distance prediction, as bounding the possible distance values means there is a maximum amount of dissimilarity. This is especially problematic for inference, where the expressiveness of the model could suffer if inputs that are far apart always result in a distance of 1. In an additional experiment, we determined that the model predicts a distance greater than 1 in 8.2% of all cases, when evaluated over more than 3000 random samples from the validation and test data sets. Thus, a significant fraction of these samples makes use of the enlarged output range.
>
> 2. Random network on TID dataset
> Correct, the random network should be bold in Table 1 as it is the best performing metric. This was a mistake on our side and will be fixed for the next revision.
>
> 3. Parameter inference for PDE models
> This is a very good point, parameter inference for PDE models w.r.t. real world data and observations is an exciting future area of application for our metric. E.g., this could be done by optimizing through a differentiable PDE solver to match experimental data. This poses a challenging learning task, as the resulting energy landscapes with regular norms like L2 can be very noisy and hard to optimize. We anticipate that our metric could help to provide improved gradient-based feedback, as it is less sensitive to typical effects of PDE simulations like translations or strong changes in single pixels, and by construction differentiable.
>
> 4. Applications of the metric
> Thank you for pointing this out. To accommodate the 8-page limit of our submission, we admittedly gave only few details here. We’d be happy to extend the discussion of possible applications in a future revision. Apart from the use case of parameter inference for experiments and observations (as mentioned above), the metric could be used as a robust and fast accuracy assessment for existing and new simulation methods. From our experience, the changes in the perturbed solutions from our data generation process are close to changes induced by different numerical approximation errors. As such, our method could guide the development of future numerical schemes by comparing competing methods to a reference solution obtained with a higher-order scheme. We are also planning to extend the LNSM metric to higher dimensional data sets where natural image metrics would not be applicable any more. We will include this discussion in the revised version of our paper.

---

### Official Review · AnonReviewer1 · 2019-10-25
**Official Blind Review #1**

**Rating:** 6

**Review:**

This paper is very well written and easy to understand. They focus no domains of data where there exists some controllable parameter(s) for data generation, using this parameter in a way that resembles self-supervised learning losses. The main contribution I think is in the use of correlations of changes in the scoring space (d) with changes in the generative parameters.

The main weakness of the paper though is the novelty / proper connection to self-supervised learning work. There is a large body of research that uses techniques similar to those outlined in this work, particularly in the use of a siamese-like network to ensure that representations follow some characteristic. Besides being very close to [1] (cited in this work), there are a number of works that try to make sure that the output (loosely, and not directly) corresponds to a binary variable indicating whether the two inputs come from the same image / sequence / graph / etc [2, 3]. Closer to this work, there are self-supervised methods that learn by "ego motion" [4] and video representation comparisons [5], to name a few.

Full disclosure: I didn't read the appendix, and there seems to be interesting / useful results there. Perhaps some of the architecture description can be scaled down in the main text and some of the useful stuff from the appendix could be added.

[1] The Unreasonable Effectiveness of Deep Features as a Perceptual Metric
[2] Data-Efficient Image Recognition with Contrastive Predictive Coding
[3] Learning Representations by Maximizing Mutual Information Across Views
[4] Learning to See by Moving
[5] Unsupervised Learning of Visual Representations using Videos

Update:
I have read the responses and generally I am happy with them. I'd like to wait on R2 and see how their concerns over the metric play out, but I'm still leaning towards acceptance.

**Experience Assessment:**

I have read many papers in this area.

**Review Assessment: Checking Correctness Of Derivations And Theory:**

I assessed the sensibility of the derivations and theory.

**Review Assessment: Checking Correctness Of Experiments:**

I assessed the sensibility of the experiments.

**Review Assessment: Thoroughness In Paper Reading:**

I made a quick assessment of this paper.

---

> ### Author Response · Authors · 2019-11-11
> **Response to Review 1**
>
> We thank reviewer 1 for the review and the insightful comments regarding other self-supervised work.
>
> It is correct that the neural network architecture of our network resembles the work of [1]. Arguably, most Siamese network architectures that rely on finding similarities in feature space share many similarities. We intentionally keep our architecture and the dimensionality of our data close to existing work to allow for meaningful comparisons to other metrics. We additionally assess a wide range of design choices for the different components of the network and provide experiments for altering them (mainly in Appendix B).
>
> Compared to the work of [1], we found that our modified training and data generation approaches lead to an improvement of 0.05 in terms of of the Spearman correlation rank across all of our test data sets. Given that the improvement of [1] over a regular L2 norm is 0.10, we believe this is a significant step forward.
>
> The works [2] and [3] could be a good addition as future work to learn a more meaningful feature space representation with our base network. The self-supervised works mentioned as [4] and [5] provide interesting insights as well, but rely on the assumption of a rigid, fixed world geometry which has keypoints that can be tracked. The volatile and non-persistent behaviour of field data is very different, as even small temporal changes typically deform and modify possible keypoint features over time. As a result, our network has to learn to extract general information from the data. This is possible as our network only has to predict a scalar distance value from both inputs, while in [4] and [5] the networks need more specialization due to the clearly more complex prediction output.

---

### Author Response · Authors · 2019-11-11
**General Response**

We would like to thank all the reviewers for their invaluable comments, questions, and suggestions. Here, we will address some common concerns from all reviews and describe changes we are planning to include in our revision.

We agree with the reviewers that it is important to more clearly point out the use cases of our metric. Specifically, the following areas could benefit from our pre-trained LNSM networks: parameter inference to match PDE models with experimental data or real-world observations, the development and assessment of new numerical schemes, and generative models for physical simulations (e.g., using GAN architectures). For the latter, our model could take the place of perceptual loss terms for natural images, where VGG is currently widely used. As PDE-based formulations and numerical simulations are used in a wide range of physical sciences, our method could have a positive impact for many disciplines.

We give further details in the individual replies below. In our revised document, which will be available within the next few days, we will include:
1. An extended description of use cases for a metric targeting PDE data
2. A discussion to clearly indicate the differences and similarities to existing self supervised learning work
3. An explanation regarding the balance of the MSE and correlation terms of our loss formulation

---

### Decision · Program_Chairs · 2019-12-19

**Decision:**

Reject

**Comment:**

The authors present a Siamese neural net architecture for learning similarities among field data generated by numerical simulations of partial differential equations. The goal would be to find which two field data are more similar to each. One use case mentioned is the debugging of new numerical simulators, by comparing them with existing ones.

The reviewers had mixed opinions on the paper. I agree with a negative comment of all three reviewers that the paper lacks a bit on the originality of the technique and the justification of the new loss proposed, as well as the fact that no strong explicit real world use case was given. I find this problematic especially given that similarities of solutions to PDEs is not a mainstream topic of the conference. Hence a good real world example use of the method would be more convincing.